# Diffusion-based generative AI for exploring transition states from 2D molecular graphs

Seonghwan Kim [1,3], Jeheon Woo [1,3] & Woo Youn Kim [1,2] ✉

The exploration of transition state (TS) geometries is crucial for elucidating chemical reaction mechanisms and modeling their kinetics. Recently, machine learning (ML) models have shown remarkable performance for prediction of TS geometries. However, they require 3D conformations of reactants and products often with their appropriate orientations as input, which demands substantial efforts and computational cost. Here, we propose a generative approach based on the stochastic diffusion method, namely TSDiff, for prediction of TS geometries just from 2D molecular graphs. TSDiff outperforms the existing ML models with 3D geometries in terms of both accuracy and efficiency. Moreover, it enables to sample various TS conformations, because it learns the distribution of TS geometries for diverse reactions in training. Thus, TSDiff finds more favorable reaction pathways with lower barrier heights than those in the reference database. These results demonstrate that TSDiff shows promising potential for an efficient and reliable TS exploration.

A transition state (TS) refers to a transient molecular configuration that places on top of the energy barrier that reactants pass through the minimum energy path to reach products, corresponding to the saddle point on the potential energy surface (PES). Identifying TSs is an important task in chemical reaction analysis, such as kinetics modeling[1–4], mechanism studies[5–12], and catalyst design[13–16]. Although TS geometries are difficult to observe experimentally due to their transient nature, they can be obtained using quantum chemical calculation methods. Over the past decades, a variety of TS optimization techniques have been developed and applied to many chemical reactions, thereby providing insights into diverse chemical phenomena[16–22].

TS optimization methods have two primary categories: single-ended[20–22] and double-ended methods[19,23–27] depending on input types. The former relies on a single set of the 3D geometries of reactants or estimated TSs. One example is the Berny algorithm[20] which optimizes a given TS guess geometry to the saddle point of the PES using the local surface information of atomic forces and a Hessian. Most single-ended approaches start from the 3D geometries of reactants, such as artificial force-induced reaction (AFIR)[28], anharmonic downward distortion following (ADDF)[29], and single-ended growing string methods (GSMs)[19]. The double-ended methods utilize the 3D geometries of both reactants and products. For example, the nudged elastic band[23] and double-ended

GSMs[24–26] first search the minimum energy pathway connecting the reactants and products and then identify the maximum energy point on that pathway. While these conventional methods are widely used in practice, they entail large computational cost and often convergence issues, making TS exploration a considerably demanding task.

Recently, there has been a growing interest in using machine learning (ML) methods to investigate the TSs, with the aim of mitigating the high cost of conventional methods. For example, numerous studies have been conducted to directly estimate barrier heights[4,30–36]. However, we here focus on the prediction of TS geometries[32,37–41], since it provides atomistic insights into reaction mechanisms and allows the refinement and validation of the predicted TS via post quantum chemical calculations. In the past few years, several ML models have been proposed to accurately predict TS geometries by leveraging the 3D geometries of reactants and products as input, like the double-ended methods[32,37–41]. The validity of these models was demonstrated with density functional theory (DFT) calculations. Meanwhile, as a concurrent work of this study, Duan et al. developed a diffusion model, OA-ReactDiff[42], to predict the highest energy image of the DFT-based climbing image NEB, also with the double-ended approach using the reactant and product geometries. These existing models have exhibited promising results on a general gas-phase reaction database[43] as well as specific reaction

[1]Department of Chemistry, KAIST, 291 Daehak-ro, Yuseong-gu 34141 Daejeon, Republic of Korea. [2]AI Institute, KAIST, 291 Daehak-ro, Yuseong-gu 34141 Daejeon, Republic of Korea. [3]These authors contributed equally: Seonghwan Kim, Jeheon Woo. ✉e-mail: wooyoun@kaist.ac.kr

categories, such as $S_N2$ and hydrogen transfer reactions, indicating their potential to complement expensive quantum chemical calculations. Despite their remarkable achievements, it should be noted that they still require well-aligned reactant and product geometries along the reaction coordinates.

Both the conventional and ML approaches need an appropriate input preparation for 3D molecular geometries. However, it is well known that the results of the conventional approaches are sensitive to the input structures[25,26,44,45]. The ML approaches also take the 3D conformations of reactants and products as input. Thus, it is inevitable for them to share the same input sensitivity issue. As pointed out by numerous studies, ML models using 3D molecular geometries as input are known to be input sensitive across various fields[31,46,47]. In the TS prediction task, Choi[40] shows that perturbations to the input geometries can result in a different TS geometry. Therefore, in practical applications, the input preparation becomes an important procedure affecting the quality of prediction results. To obtain an appropriate input, the molecular orientation should be considered along the target reaction coordinates, which requires careful consideration even by professional chemists. Moreover, it is necessary to explore possible TS conformations to elucidate the most favorable reaction pathway[48–50], which makes the preparation task more demanding to cope with various conformations.

To address this problem, we present TSDiff, an ML model that learns a direct mapping between TS conformations and 2D molecular graphs. Thus, one can skip the proper selection of conformations and orientations. Moreover, TSDiff can generate various TS conformations possible from the 2D graph with high reliability by employing the stochastic diffusion method which has been used to generate molecular conformers in equilibrium[51–53]. Consequently, TSDiff can minimize user efforts throughout the entire TS generation process and explore multiple reaction pathways without the direct consideration of conformations, leading to high efficiency.

In this study, the performance of TSDiff was evaluated using Grambow's dataset[43], a set of diverse gas-phase organic reactions generated with the single-ended GSM, where reactant molecules were sampled from GDB-7 to cover reactions involving possible bond changes among C, H, O, and N atoms. Despite its simplified input of 2D graphs, TSDiff has achieved the highest accuracy compared to the existing methods that rely on 3D geometric information. The validity of the multiple TS conformations generated by TSDiff was verified by quantum chemical calculations based on DFT. First, saddle point optimization[20] was performed on the generated geometries to obtain the TS geometries with a single imaginary vibrational frequency. The intrinsic reaction coordinate (IRC) calculation[54] was followed to validate that TS geometries correspond to the given graphically defined reaction. The detailed validation methodology is provided in the Computational details section. TSDiff achieved a significantly high success rate of 90.6% in this validation, showing its reliability as an initial TS geometry guesser. Based on these results, we expect that TSDiff can greatly alleviate the time-consuming trial-and-error procedures of TS exploration. We also found 2303 new TS conformations at saddle points other than the reference using TSDiff with eight rounds of sampling for 1197 reactions in the test set. Some of these corresponded to lower barrier heights than those of the references, suggesting more favorable reaction pathways. It is worth noting that TSDiff was trained with only one TS conformation for each reaction, underscoring its generative power in this context. Overall, our findings demonstrate the potential of TSDiff as a promising approach for efficient and reliable TS exploration.

## Results

### A brief description of the generation process
In this section, we provide a brief overview of TSDiff, an ML model designed to learn the conditional distribution of 3D TS geometries given 2D reaction information presented as SMARTS[55] (see Fig. 1a).

TSDiff is based on the stochastic denoising diffusion method, where the model is trained to learn the reverse process of a noise process that adds a random noise to the given geometry at each discrete time step. At the inference phase, TS geometries are generated from an initial state with a complete noise through the iterative denoising process, where the noisy input is gradually refined by the denoising neural network at each time step, given the 2D reaction information (see Fig. 1b).

The input of the model is 2D reaction information expressed as a reaction graph $\mathcal{G}_{rxn}$ which captures the bond changes in reactants and products[56]. The simplified version of the reaction graph is depicted in the left box of Fig. 1a. Molecular graphs for reactants and products, $\mathcal{G}_R$ and $\mathcal{G}_P$, can be constructed based on bond and atom information that can be obtained from SMILES[57]. The nodes in the graph are represented as atom-feature vectors containing atomic numbers. For the edges, the molecular graphs utilize extended graph edges that include node-pair indices within a 3-hop graph distance in the raw graph created based on covalent bonds. The condensed reaction graph, which serves as our model input, is formed by combining the two graphs of reactants and products using atom-mapping information.

TSDiff employs graph neural network (GNN) layers based on SchNet[58] for its denoising neural network to handle the noisy positions and reaction graphs. The construction of a geometric reaction graph involves adding the noisy positions to the 2D reaction graph and connecting nodes with interatomic distances smaller than a specified cutoff radius. This process integrates bond information, graph distance information, and spatial distance information as edge-features in the geometric graph. Subsequently, the model leverages these geometric reaction graphs to approximate a score function, a gradient of log-likelihood for noisy TS conformations, which is applied to denoising by updating the noisy positions toward the correct TS geometry. More details are described in the TSDiff section.

The proposed TSDiff was trained and validated using the general organic gas-phase reaction database published by Grambow et al.[43]. We employed an ensemble with a total of eight models, and our training process took 22 h for each model on a single RTX 2080 Ti NVIDIA GPU. For most results reported in the following sections, we used the ensemble model and will refer to it as TSDiff unless otherwise noted. Diffusion models entail higher inference costs compared to other deep learning models, mainly due to their iterative denoising process. Specifically, TSDiff requires 5000 denoising steps for inference, which takes a few seconds per reaction. However, this cost is negligible compared to DFT calculations for TS optimization.

### Generation of TS conformations
We emphasize that TSDiff is a stochastic generative model, implying that different geometries are generated at each sampling. Figure 2 depicts a conceptual representation of TSDiff's predictive distribution. The different geometries generated by TSDiff correspond to specific TS conformations that can be built from the same 2D reaction graph. For example, Fig. 3 shows several generated geometries corresponding to specific conformations and reference geometries for three reactions in the test set. This is an inherent outcome because TSDiff uses only 2D graphs as input. Also, the reference TS would be one of various TS conformations identified by a specific computational method. Therefore, it is essential to consider diverse conformations and the comparative analysis of their barrier heights in the TS exploration process in order to identify the most favorable reaction pathway. Thanks to the nature of generative AI, TSDiff learns the distribution of TS geometries for diverse reactions in training, facilitating the reliable sampling of these different TS conformations.

(a)

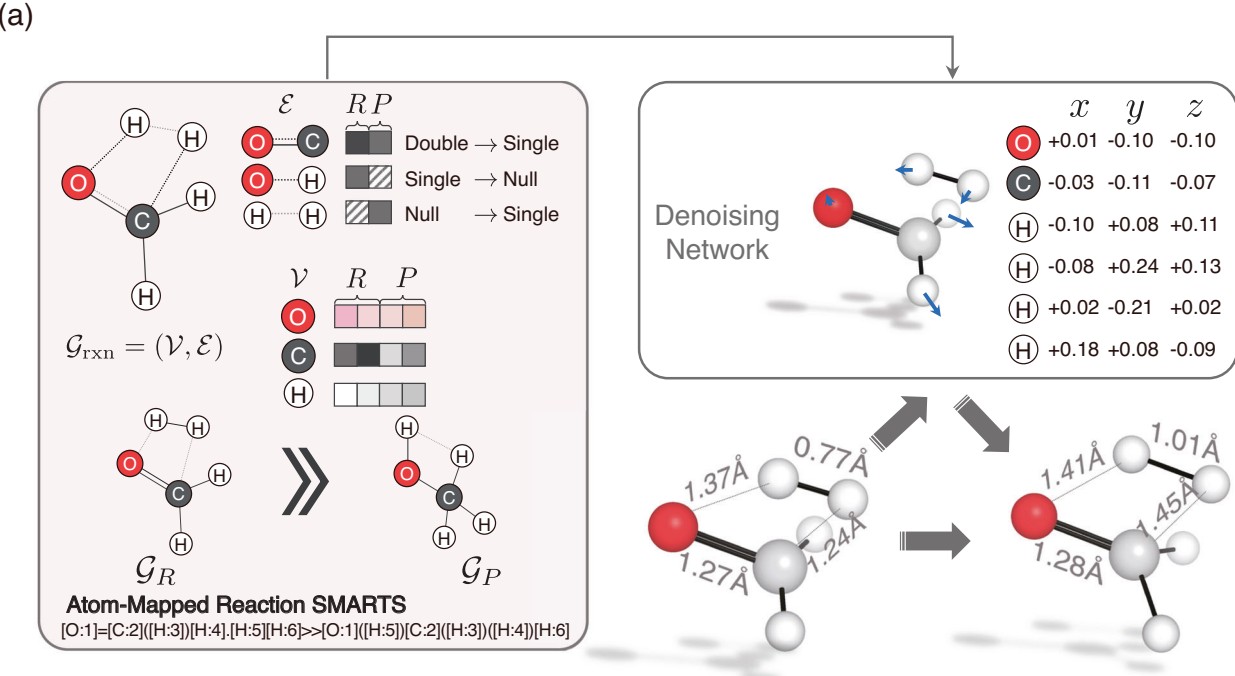

(b)

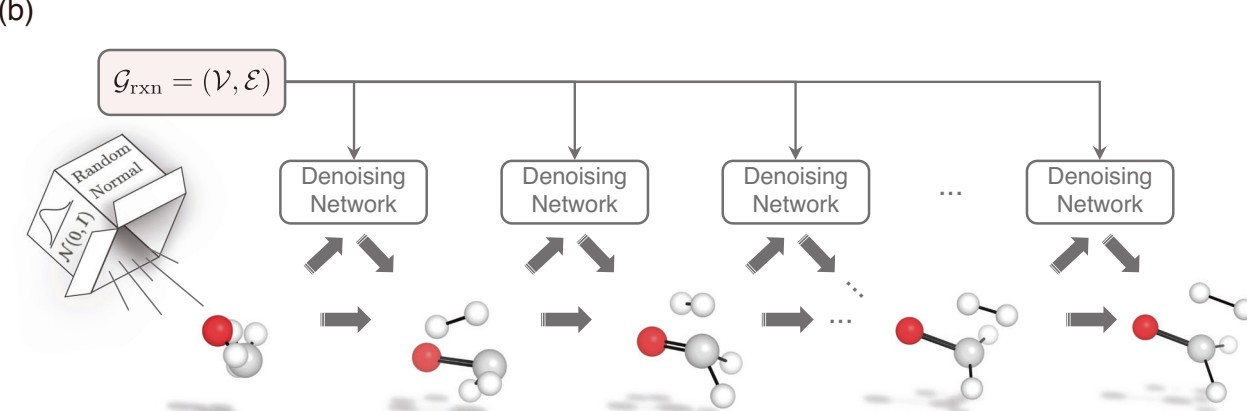

**Fig. 1 | Overview of the proposed TSDiff. a** Illustration of the reaction graph ($\mathcal{G}_{rxn}$) and the denoising network. The molecular graphs of the reactants (*R*) and products (*P*), denoted $\mathcal{G}_R$ and $\mathcal{G}_P$ respectively, are constructed from the SMARTS representation of the reaction. Then, the condensed graph $\mathcal{G}_{rxn}$ is formed using the node vectors ($\mathcal{V}$) and edge vectors ($\mathcal{E}$) obtained from them. The denoising network denoises a given geometry input based on $\mathcal{G}_{rxn}$. **b** Transition state (TS) generation procedure of the proposed TSDiff. Starting from a randomly initialized geometry, the geometry is progressively refined by the denoising network until reaching a predicted TS geometry. All molecular geometries were plotted using PyMOL[71].

The various conformations generated by TSDiff need to be verified to ensure that they are indeed chemically valid TSs. As an example, we performed quantum chemical validation on a single test reaction. First, TSDiff generated one hundred samples for this reaction, which were then optimized using a saddle point optimization. Figure 4 visualizes the distribution of the generated geometries using t-distributed stochastic neighbor embedding (t-SNE)[59] in scikit-learn[60]. Each dot in the figure represents a generated geometry, while the star and cross-shaped dots indicate the optimized and reference geometries, respectively. In addition, each dot has been color-coded to reflect its respective optimization result. For example, all generated geometries represented by the blue dots were optimized to the geometry represented by the blue star-shaped dot via the saddle point optimization.

All one hundred generated geometries were successfully optimized to saddle points, resulting in nine different TS conformations. The images on the right side of Fig. 4 show the optimized conformations. On the t-SNE projection map, it is evident that similar

conformations tend to cluster together and be closely located to their respective optimized results. This character of the generated samples suggests that an efficient search for TS conformations is possible without having to perform quantum chemical calculations on the entire generated samples. Many clustering algorithms are already available, offering an effective means to select representative conformation samples. We also present an illustrative experiment in the Supplementary Discussion, showcasing the practical application of a clustering algorithm in TS exploration using TSDiff.

The nine different conformations were caused by two rotatable single bonds closest to oxygen, C–C and C–O. There were three major conformational changes with different dihedral angles centered on the C–C bond, and for each there were three minor conformations with different dihedral angles centered on the C–O bond, resulting in a total of nine different conformations. Here, the accuracy of the generated geometries was measured by the mean absolute error (MAE) of interatomic distances, D–MAE, with respect to their corresponding

optimized results. A detailed method for measuring the D−MAE is given in the Measurement details section. The resulting average D−MAE was very small at 0.045 Å, indicating that the generated samples were optimized to the respective saddle points with only minor adjustments.

The resulting optimized geometries were all confirmed as valid TS geometries using IRC calculations. A detailed description of the IRC and validation method is provided in the Computational details section. This result indicates that TSDiff can find not only a conformation that corresponds to the reference geometry but also other valid TS conformations. Interestingly, the discovered TS conformations include TSs with lower barrier heights than that of the reference, as shown in Fig. 4, where the barrier heights were calculated with the energy difference between the optimized TSs and the reference

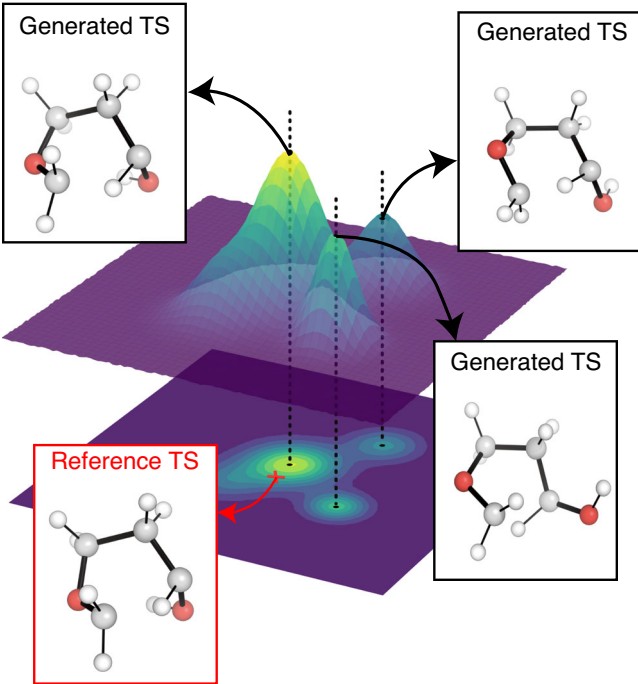

**Fig. 2 | Conceptual illustration of predictive distribution of TSDiff.** The 3D surface represents a probability distribution of predicted transition state (TS) geometries. Examples of reference and generated TS geometries are visualized in the boxes. The generated geometries extracted from each peak have different conformations from each other. The reference TS is marked by a red cross positioned on the probability distribution contour. All molecular geometries were plotted using PyMOL[71].

reactants. This proves that the reference TS may not be the most favorable one, even though it was built using the most stable molecular conformation of the reactants[43], highlighting the importance of exploring multiple TS conformations. As a result, we demonstrate that the stochastic diffusion model, which has already shown its capability of accurately generating conformers in equilibrium, can be extended to TS explorations.

## Performance of TS generation

From the perspective of a generative AI, it is important to evaluate the ability of TSDiff to generate samples that cover the reference TS of the dataset and how accurate the generated samples are. To this end, we calculated the following two metrics for all reactions in the test set: coverage (COV) and matching (MAT) scores. The COV score measures the percentage of reference TS geometries covered by the predicted ones by TSDiff, where a reference is considered to be covered if there exists any predicted one having a D−MAE within a criterion of $\delta$ with the reference. We used two criterion values: $\delta = 0.1$ Å and $\delta = 0.2$ Å. The value of 0.1 Å was determined based on the accuracy of a state-of-the-art model[40] that has demonstrated reliability with a high success rate in quantum chemical validations. However, this can be a strict criterion, so we also evaluated a COV score of $\delta = 0.2$ Å. The MAT score measures the similarity between generated and reference samples by calculating the minimum D−MAE between the generated geometries and the reference geometry. The mathematical definitions of these two metrics can be found in the Measurement details section.

While these two metrics are widely used for generative AI evaluation, it should be noted that they have a limited application in this study because there is only one TS conformation for each reaction in the reference dataset. Therefore, for each individual reaction, the COV score is either 100% or 0% if only a single sample is used for evaluation. Thus, Table 1 presents the COV and MAT scores of TSDiff according to the number of sampling, including a comparison to the scores of TSDiff without the ensemble method. The distributions of the MAT scores across the reaction are shown in Fig. 5.

The COV and MAT scores improve rapidly as the number of sampling increases. This is because TSDiff's performance is underestimated at small numbers of the sampling, as it generates many different conformations, as shown in Fig. 3. In addition, the ensemble method led to slight performance improvements in both COV and MAT scores. As a result, TSDiff is expected to be able to generate 84.0% of the reference TSs with a D−MAE of 0.1 Å or less within ten rounds of sampling.

We compare the features and accuracies of TSDiff and the existing ML models in Table 2. To the best of our knowledge, Table 2 includes all the models whose performance has been reported on the Grambow's dataset[43]. Previous works commonly used geometries of reactants and products for their input features[37–40], and the most recently

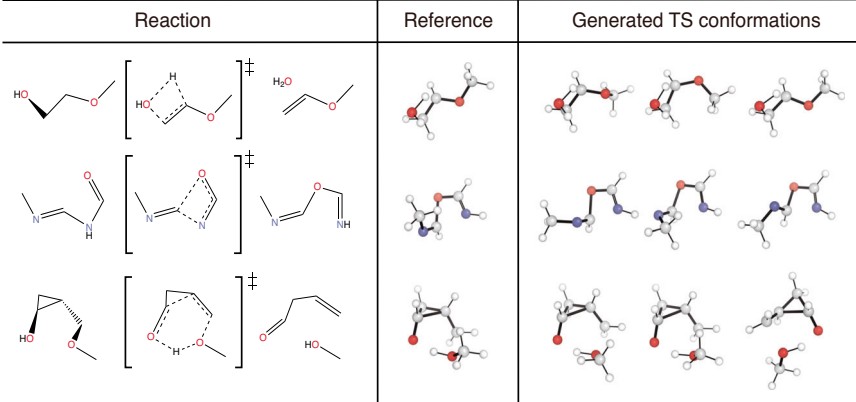

**Fig. 3 | Examples of transition state (TS) conformations generated by TSDiff.** For each reaction, we sampled multiple TS geometries and picked three different conformations including one that was best-aligned with the reference one. All molecular geometries were plotted using PyMOL[71].

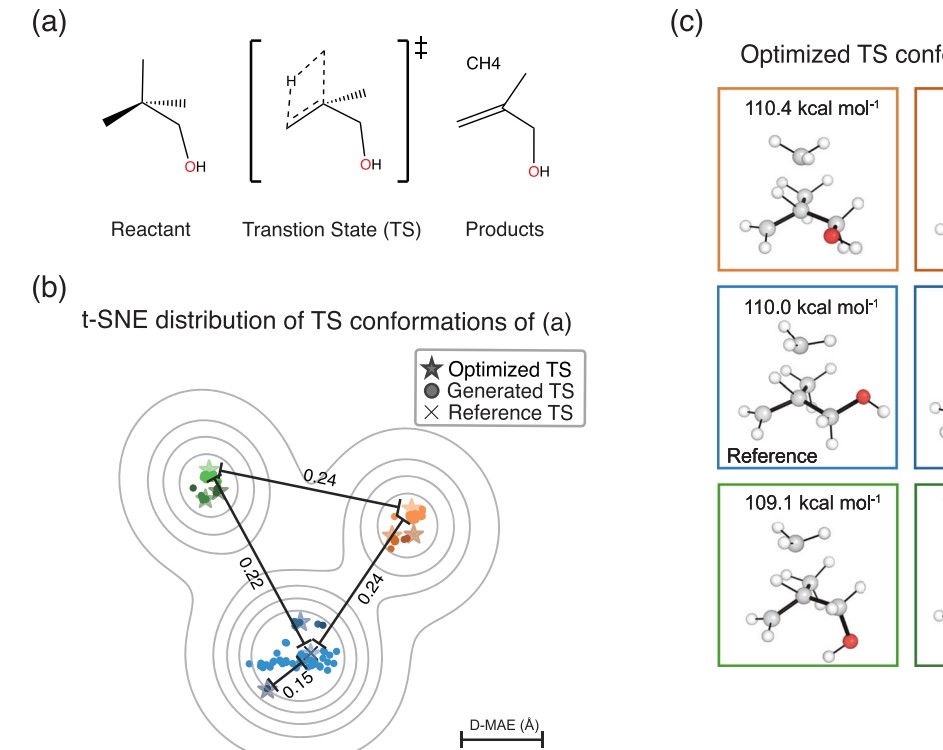

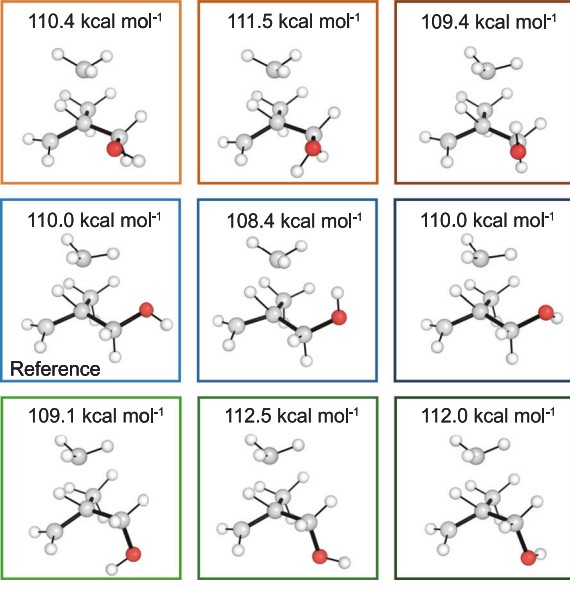

**Fig. 4 | Visualization of the geometries generated by TSDiff. a** A given chemical reaction to demonstrate the use of TSDiff. **b** Visualization of the distribution of transition state (TS) conformations of the given reaction, using the t-distributed stochastic neighbor embedding (t-SNE) method based on the interatomic distances of the geometries. The geometries generated by TSDiff and the optimized results for them are indicated by dots and star-shaped dots, respectively. The reference TSs in the dataset are marked with a cross. The four values in this plot indicate the mean absolute error of the interatomic distances (D–MAE) between the two selected dots. The contours were plotted using kernel density estimation on the t-SNE transformed results. **c** The geometries and barrier heights corresponding to the nine different TS conformations, represented by the star-shaped dots in (**b**) with the same color as the box edges. All molecular geometries were plotted using PyMOL[71].

reported model by Choi additionally used the interpolated geometries of reactants and products[40]. Their prediction targets are broadly divided into two categories: interatomic distances and atomic positions. The models that predict interatomic distances require an additional step of a nonlinear least-squares optimization to restore the distances to atomic positions. In contrast, Jackson et al. directly predicted atomic positions using tensor field networks[37]. TSDiff also targets atomic positions directly in its generation process. A unique feature that distinguishes it from the existing models is that TSDiff uses only 2D graphs as input.

Before comparing the accuracy of the models, it is important to note that evaluating the accuracy of TSDiff under the same conditions

## Table 1 | The coverage (COV) and matching (MAT) scores of TSDiff according to the number of sampling

| # of sampling | Ensemble | | | No ensemble | | |
|---|---|---|---|---|---|---|
| | COV[a]↑ | COV[b]↑ | MAT ↓ | COV[a]↑ | COV[b]↑ | MAT ↓ |
| 1 | 49.1 | 73.9 | 0.137 | 47.9 | 73.9 | 0.140 |
| 3 | 67.3 | 87.5 | 0.096 | 65.9 | 86.7 | 0.100 |
| 5 | 75.2 | 92.6 | 0.079 | 74.2 | 91.4 | 0.084 |
| 10 | 84.0 | 95.6 | 0.063 | 80.4 | 93.7 | 0.072 |
| 100 | 91.7 | 97.7 | 0.045 | 89.7 | 97.2 | 0.052 |

The results of two TSDiff models with and without the ensemble method are compared. The units for COV and MAT values are percent (%) and angstroms (Å), respectively.
[a]The COV score was calculated using the mean absolute error of interatomic distance (D–MAE) threshold of 0.1 Å.
[b]The COV score was calculated using the D–MAE threshold of 0.2 Å.

as existing models is not straightforward because TSDiff can generate other TS conformations, while we only have a single TS conformation available as a reference. To address this, we evaluated the accuracy of TSDiff on a TS conformation only if it directly matches the corresponding reference. To identify these matching samples, we conducted saddle point optimizations on the generated samples. For a reliable comparison, we aimed to find as many test reactions with a matched reference as possible, so we performed the optimization on TS conformations generated with a total of eight rounds of sampling for each reaction. Samples with a D–MAE between their optimized result and the reference of less than 0.01 Å were considered matching, resulting in the covering rates of 53.2% and 84.6% of the test reactions with one and eight sampling rounds, respectively. One of the generated samples is randomly selected to calculate the D–MAE when multiple samples match the same reference TS in a given reaction graph, which gives a consistent D–MAE value regardless of the number of sampling rounds to facilitate fair evaluation.

Table 2 shows the D–MAE values of TSDiff with and without considering conformer matching. The latter is the case for a single sampling, the resulting conformation of which can be considered an approximate TS for the respective reference. In this case, the D–MAE value is 0.137 Å, which is lower than those of all models except Choi's one, indicating that TSDiff is fairly accurate when only providing a single TS without conformer matching. Furthermore, considering the conformer matching, the D–MAE values become 0.063 Å and 0.067 Å for one and eight sampling rounds, respectively, which are considerably lower than those of all. Note that while the covering rate increased from 53.2% to 84.6%, the D–MAE value remained consistent, suggesting its reliability as a metric to assess accuracy. Thus, it can be

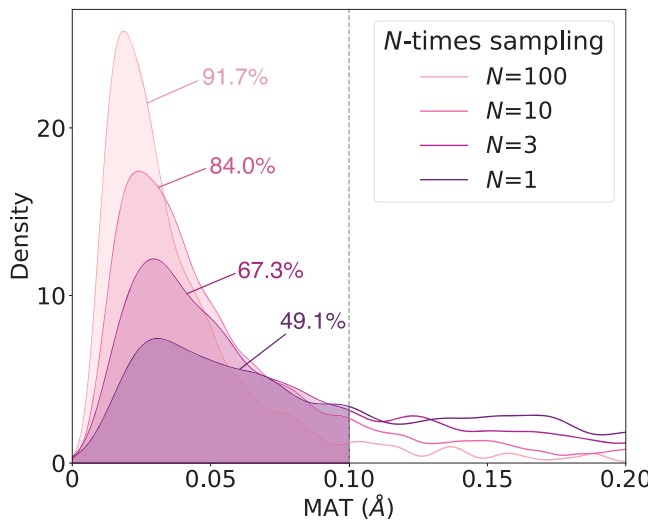

**Fig. 5 | The distribution of the matching (MAT) scores of TSDiff with the number of sampling.** The percentages represent the coverage (COV) scores measured using the mean absolute error of interatomic distances (D−MAE) threshold of 0.1 Å. The source data is provided as a Source data file.

## Table 2 | Comparison of accuracy and features between machine learning models

| Model | Input type | Target type | D−MAE (Å) |
|---|---|---|---|
| Makoś et al.[38] | $\mathcal{C}_R, \mathcal{C}_P$ | CM[a] | 0.170 |
| Jackson et al.[37] | $\mathcal{C}_R, \mathcal{C}_P$ | Positions | 0.244[b] |
| Pattanaik et al.[39] | $\mathcal{C}_R, \mathcal{C}_P, \mathcal{G}_R, \mathcal{G}_P$ | Distances | 0.225[b] |
| Choi[40] | $\mathcal{C}_R, \mathcal{C}_P, (\mathcal{C}_R+\mathcal{C}_P)/2$ | Distances | 0.095[b] |
| TSDiff | $\mathcal{G}_R, \mathcal{G}_P$ | Positions | 0.137[c], 0.063[d], 0.067[d] |

The input type, target type, and accuracy of the models are compared. $\mathcal{C}_R$ and $\mathcal{C}_P$ denote the geometries of reactants and products, respectively, and $\mathcal{G}_R$ and $\mathcal{G}_P$ denote the 2D graphs of reactants and products, respectively. $(\mathcal{C}_R+\mathcal{C}_P)/2$ denotes the interpolated geometry between $\mathcal{C}_R$ and $\mathcal{C}_P$.
[a]CM indicates the Coulomb matrix.
[b]The values are borrowed from Choi[40].
[c]The value was evaluated without considering conformer matching, meaning that a single generated transition state (TS) for each reaction was used to evaluate the mean absolute error of interatomic distance (D−MAE).
[d]These values were calculated for TSs generated from single and eight sampling rounds, only if they matched the corresponding reference geometry after saddle point optimization, covering 53.2% and 84.6% of the test reactions, respectively.

concluded that TSDiff generates TS geometries with better accuracy than the existing models, without computationally expensive 3D geometric information. Further analysis and methodology involving quantum calculations will be presented in subsequent sections.

**Quantum chemical validation of generated conformations**
As an extension of the experiment in Fig. 4, we evaluate the chemical validity of TS conformations generated by TSDiff across reactions in the test set. We performed TS optimizations based on DFT using the generated geometries for 1197 reactions in the test set. Eight geometries were generated for each reaction, and saddle point optimization was performed on the resulting 9576 geometries. Of these, 9289 were successfully optimized with a single imaginary vibrational frequency, giving a high success rate of about 97.0%, which clearly demonstrates the reliability of TSDiff as an efficient initial TS guesser. To determine how various TS conformations were found by TSDiff, we counted the number of differently optimized results. The optimized geometries were distinguished if the D−MAE between them was greater than 0.01 Å. As a result, we confirmed that 3316 unique TS conformations were

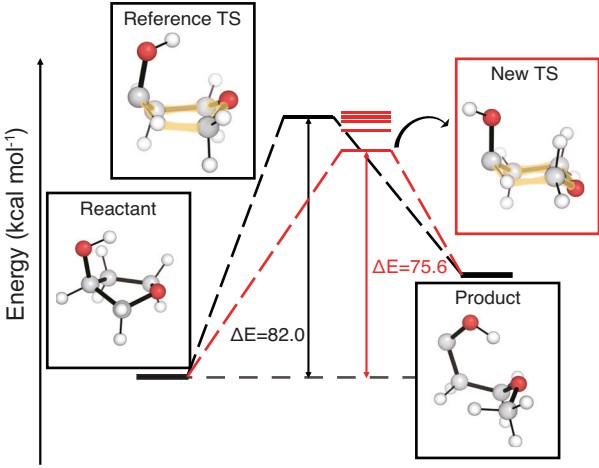

**Fig. 6 | Comparison of the barrier heights of the reference and those found by TSDiff.** The transition state (TS) conformations were obtained by saddle point optimization using the initial geometries generated by TSDiff, and their energy levels are shown as red lines. The geometries of the reference database are visualized in the black boxes, and the newly discovered TS conformation with the lowest energy is visualized in the red box. The barrier heights ($\Delta E$) are compared for the reference TS and the new lowest energy TS. All molecular geometries were plotted using PyMOL[71].

obtained, of which 2303 samples corresponded to different saddle points than those of the reference TSs.

The IRC validation was carried out to verify that the optimized TS geometries are on the reaction pathways connecting the correct reactants and products. This verification process enables the validation of the precision of TSDiff. Due to the huge computational cost, the IRC calculations were performed on one randomly selected geometry for each reaction. For 998 reactions, corresponding to 83.4% of the test set, the IRC calculations successfully converged by linking the optimized geometries to the correct reactants and products, demonstrating the high precision of the TSDiff model.

In our investigation of the energetics of the successfully validated TSs through IRC, we discovered 309 new TSs with energies lower than those of the reference TSs by more than 0.1 kcal mol$^{-1}$. Moreover, after conducting further geometry optimization on the reactants obtained from the IRC calculation, we identified 513 pathways with barrier heights lower than those of the reference. The increase from 309 to 513 is attributed to the higher energy of the newly obtained reactants than the reference reactants, which is traced back to the inclusion of the conformational search for reactants in the generation of the reference data. These findings imply that lower equilibrium geometries do not always correspond to reaction pathways with the lowest overall TS barriers.

Figure 6 shows an example case where TSDiff found TS conformations with lower energies than that of the reference. Five different TS conformations were obtained by saddle point optimization, and their energy levels are indicated by the red lines. The lowest energy conformation in the red box has an energy 6.4 kcal mol$^{-1}$ lower than that of the reference, suggesting a more favorable reaction pathway. This substantial energy difference between the reference and the new TS conformations is due to the apparent geometric difference between them, where the hexagonal rings, highlighted in yellow, have the boat and chair conformations in the reference and the new TS, respectively. This emphasizes the importance of searching for different TS conformations to find a more favorable reaction pathway, and TSDiff can be used for this purpose. The results of the reaction conformational search utilizing TSDiff for more complex reactions can be found in the section Analysis on multiple reaction pathways explored by TSDiff.

## Analysis on failure cases

To further assess the coverage ability of TSDiff based on DFT calculations, we analyzed 199 reactions where the model failed to generate the correct TS within a single sampling. We performed up to four additional random samplings and validated the generated samples using the same process as in the previous section. This resulted in the identification of 89 correct TS geometries among 199. Of the remaining 110 reactions still not covered by TSDiff, we found that several samples were successfully optimized to their reference geometries but failed the IRC validation. We note that the IRC calculation of the reference TS may fail due to the lack of an IRC verification step during the data generation process[40,43].

Considering this point, we conducted IRC calculations on the reference TSs of the 110 reactions to confirm whether or not they are elementary step reactions. Among them, 95 reference TSs failed the IRC calculation, which highlights requirements of re-evaluating TSDiff's performance on the remaining 1102 reactions after excluding them. Then, TSDiff achieved a success rate of 97.4%, with 8588 of the 8816 samples successfully optimized to the saddle point, as described in Table 3. The success rate of IRC validation for geometries generated by the single sampling was also recalculated to 90.6%, which was reported as 83.4% in the previous section. Finally, 98.6% of the refined test reactions were successfully covered by TSDiff within five rounds of sampling, which resulted in the correct TSs on 1087 reactions.

## Analysis on multiple reaction pathways explored by TSDiff

We present additional results for four reactions in which TSDiff discovered TS conformations with distinct reaction coordinates. For each reaction, two TS conformations were obtained by saddle point optimization of the generated TSs, resulting in D–MAE values between two conformations of 0.29 Å, 0.30 Å, 0.26 Å, and 0.15 Å for the four reactions, respectively. Subsequently, we obtained the energy profiles of the respective reaction pathways using IRC calculations. As a result, we identified two different reaction pathways for each reaction, despite both pathways sharing the same reaction graph. Figure 7 illustrates the IRC energy profiles of the reaction pathways along with the corresponding reactant, TS, and product geometries, clearly demonstrating the distinct reaction coordinates.

The reaction pathways depicted in Fig. 7a–c exhibit different bond breaking/formation sequences. For instance, Fig. 7a displays two reaction pathways with hydrogen molecules approaching from different directions in multimolecular reactants, highlighting the effectiveness of TSDiff in capturing TSs without considering alignments. In Fig. 7b, the top reaction involves cleavage of the C–C bond followed by the C–N bond, while the sequence is reversed in the bottom reaction. This difference is also evident in the bond lengths in the TS conformations shown in the figure. Similarly, in Fig. 7c, the top reaction involves cleavage of the C–O bond followed by the O–H bond, whereas the sequence is reversed in the bottom reaction. Furthermore, we observed the reaction with identical reactant and product, but involving different reactive atoms, as illustrated in Fig. 7d. In this reaction, a different hydrogen atom migrates to the neighboring carbon. For a clearer visualization, we have colored the migrating hydrogen atom in orange.

## Discussion

One of the main advantages of TSDiff is its ability to find TSs without considering the conformations of the reactants and products and their alignments. Since TSDiff does not rely on specific conformations, it allows efficient exploration of TSs in graphically defined reactions with a more generalized approach. We demonstrated its usefulness in chemical reaction analysis by generating diverse, high-quality TSs based on a given molecular connectivity. This is amazing

**Table 3 | Success rates of quantum chemical validations on geometries generated by TSDiff**

| Test reaction | # of reactions | Saddle point | IRC |
|---|---|---|---|
| Original | 1197 | 97.0% | 83.4% |
| Refined | 1102 | 97.4% | 90.6% |
| Refined | 1102 | 99.9%[a] | 98.5%[a] |

For a total of 1197 reactions, the saddle point optimization was performed on the transition state (TS) geometries generated with eight rounds of sampling, while the intrinsic reaction coordinate (IRC) calculation was performed on one randomly selected geometry for each reaction. We found 95 invalid reactions in the original test set for which the reference TS failed the IRC validation. Therefore, the success rates of the quantum chemical calculations were recalculated with the refined 1102 reactions, after excluding the failed ones.
[a]The value is the cumulative success rate over five rounds of sampling.

considering that TSDiff learned only one TS conformation for each reaction during the training phase. TSDiff was able to effectively capture TS conformations resulting from rotatable bonds in non-reactive coordinates and different reaction coordinates. Furthermore, TSDiff's transferability is supported by its successful application to another benchmark dataset, as described in the Supplementary Discussion. Here, TSDiff also proves to be an effective initial TS guesser, requiring only a small number of force calls during subsequent TS optimization. Therefore, this study shows the promising potential of TSDiff for efficient and reliable TS exploration.

These findings show that the stochastic diffusion method proven to accurately create diverse conformers in equilibrium states can be extended to TS explorations. However, it is important to recognize a limitation of this work, particularly its current restriction to organic reactions. Although inorganic databases exist, such as the FH51 set in the GMTKN55 database[61], which contains 51 reactions in small inorganic and organic systems, and another database containing about 400 reactions[38] including transition metals, the lack of large inorganic reaction databases limits the applicability of machine learning approaches in this domain. Nevertheless, with the ongoing accumulation of data in the future, we anticipate that the utility of TSDiff will expand to encompass a broader range of chemical reactions, including those involving inorganic species.

## Methods
### TSDiff

The diffusion process is a stochastic process where atomic positions change into chaotic states at discrete time steps. For a reference TS geometry $\mathcal{C}_0$ and a geometry at time step $t$ of the diffusive process $\mathcal{C}_t$, we define the probability distribution $q$ of the diffusive process as follows:

$$q(\mathcal{C}_t|\mathcal{C}_{t-1}) = \mathcal{N}(\mathcal{C}_t; \sqrt{\alpha_t}\mathcal{C}_{t-1}, \beta_t I),$$
$$q(\mathcal{C}_t|\mathcal{C}_0) = \mathcal{N}(\mathcal{C}_t; \sqrt{\bar{\alpha}_t}\mathcal{C}_0, (1-\bar{\alpha}_t)I), \quad (1)$$

where the hyperparameters $\beta_t$ and $\alpha_t(=1-\beta_t)$ denote the noise and signal schedulers, respectively, which determine how much noise to add and how much of the existing signal to preserve at time step $t$, and $\bar{\alpha}_t = \prod_{s=1}^{t} \alpha_s$. The neural network model learns a distribution parameterized by $\theta$, $p_\theta$, that simulates the reverse process of the diffusion process:

$$p_\theta(\mathcal{C}_{t-1}|\mathcal{C}_t, \mathcal{G}_{rxn}) = \mathcal{N}(\mathcal{C}_{t-1}; \mu_\theta(\mathcal{C}_t, \mathcal{G}_{rxn}, t), \sigma_t^2 I), \quad (2)$$

where $\mathcal{G}_{rxn}$ denotes the input reaction graph, and $\mu_\theta$ and $\sigma_t^2$ denote the mean and the variance of the distribution, respectively. The loss form is defined as the KL divergence between the posterior of $q$ and $p_\theta$ at

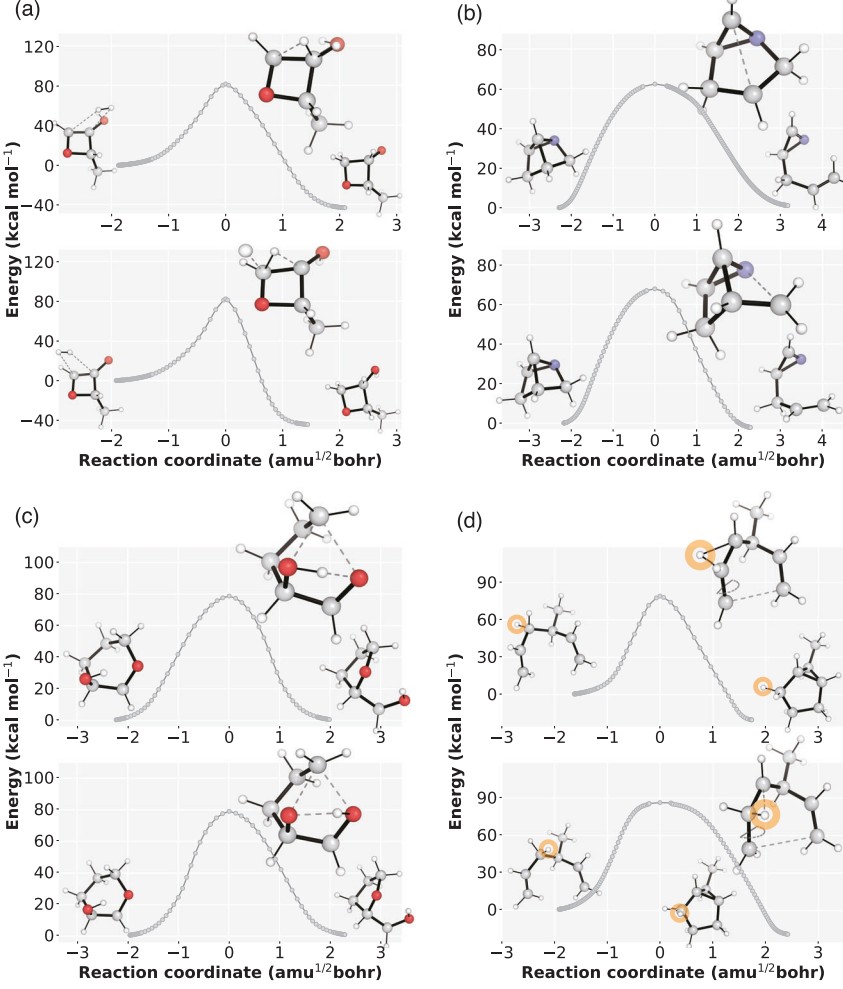

**Fig. 7 | Intrinsic reaction coordinate (IRC) energy profile of four reactions.** The energy profiles were obtained by IRC calculation of optimized transition states (TSs) based on the geometries generated by TSDiff. The points on the plots represent the IRC points, which are then connected by line segments. For each reaction, two reaction pathways were plotted, along with the corresponding geometries of the reactants, TS, and products. All molecular geometries were plotted using PyMOL[71]. The source data is provided as a Source data file.

each time step $t \leq T$:

$$
\begin{aligned}
\log p_\theta(\mathcal{C}_0|\mathcal{G}_{\mathrm{rxn}}) \geq &-\sum_{t=2}^{T} \mathrm{KL}(q(\mathcal{C}_{t-1}|\mathcal{C}_t,\mathcal{C}_0) \parallel p_\theta(\mathcal{C}_{t-1}|\mathcal{C}_t,\mathcal{G}_{\mathrm{rxn}})) \\
&- \mathrm{KL}\big(q(\mathcal{C}_T|\mathcal{C}_0) \parallel p_\theta(\mathcal{C}_T)\big) \\
&+ \mathbb{E}_{q(\mathcal{C}_1|\mathcal{C}_0)}\big[\log p_\theta(\mathcal{C}_0|\mathcal{C}_1,\mathcal{G}_{\mathrm{rxn}})\big],
\end{aligned}
\tag{3}
$$

where $p_\theta(\mathcal{C}_T)$ is a unit Gaussian prior. Minimizing the KL divergence of Eq. (3) implicitly maximizes $\log p_\theta(\mathcal{C}_0|\mathcal{G}_{\mathrm{rxn}})$ which is the main objective of the reference TS generation.

From the Gaussian distribution assumption of $p_\theta$ and $q$, the KL term is derived to the simple form as:

$$
\mathrm{KL}\big(q(\mathcal{C}_{t-1}|\mathcal{C}_t,\mathcal{C}_0) \parallel p_\theta(\mathcal{C}_{t-1}|\mathcal{C}_t,\mathcal{G}_{\mathrm{rxn}})\big) = \mathbb{E}_q\left[\frac{\beta_t^2}{2\alpha_t(1-\bar{\alpha}_t)\sigma_t^2}\left|\varepsilon - \varepsilon_\theta(\mathcal{C}_t,\mathcal{G}_{\mathrm{rxn}},t)\right|_2^2\right],
\tag{4}
$$

where $\varepsilon = \frac{\mathcal{C}_t - \sqrt{\bar{\alpha}_t}\mathcal{C}_0}{\sqrt{1-\bar{\alpha}_t}} = -\sqrt{1-\bar{\alpha}_t}\nabla_{\mathcal{C}_t}\log q(\mathcal{C}_t|\mathcal{C}_0)$ and $\varepsilon_\theta = \frac{\sqrt{1-\bar{\alpha}_t}}{\beta_t}\left(\mathcal{C}_t - \sqrt{\bar{\alpha}_t}\mu_\theta\right)^{62}$. The tractable loss form of Eq. (4), $\parallel \varepsilon - \varepsilon_\theta \parallel_2^2$, can be interpreted as the score matching loss[51,63,64]. In practice, all coefficient terms in Eq. (4) were set to 1, following the previous study[62].

The distribution $p_\theta(\mathcal{C}_0)$ should be SE(3) invariant. To ensure this, each backward transition is required to follow an SE(3) equivariant distribution, which can be addressed by utilizing the pairwise distances $\mathbf{d} = \{d_{ij}\}_{(i,j)\in\mathcal{E}}$. According to Shi et al.[65], the equivariance of $\nabla_{\mathcal{C}_t}\log q(\mathcal{C}_t|\cdot)$ can be addressed by decomposing it into $\frac{\partial\mathbf{d}_t}{\partial\mathcal{C}_t}\nabla_{\mathbf{d}_t}\log q(\mathbf{d}_t|\cdot)$. This ensures that the score function calculated in the distance coordinate is SE(3) invariant, and the partial derivative term is SE(3) equivariant. By assuming $\nabla_{\mathbf{d}_t}\log q(\mathbf{d}_t|\mathbf{d}_0) = -\frac{\mathbf{d}_t - \sqrt{\bar{\alpha}_t}\mathbf{d}_0}{1-\bar{\alpha}_t}$, the approximation target $\varepsilon$ is re-formulated as $\frac{\partial\mathbf{d}_t}{\partial\mathcal{C}_t}\frac{\mathbf{d}_t - \sqrt{\bar{\alpha}_t}\mathbf{d}_0}{\sqrt{1-\bar{\alpha}_t}}$. Accordingly, the neural network model was designed to predict the score function in distance coordinate, which is then converted to the Euclidean coordinate by applying $\frac{\partial\mathbf{d}_t}{\partial\mathcal{C}_t}$.

TSDiff employed a total of seven modified SchNet layers. A geometric reaction graph is constructed by adding the noised positions to the 2D reaction graph, and it is fed to the GNN layers. The node-vector update step is a conventional message-passing process defined as

$$
\mathbf{h}_i^{l+1} = \mathrm{MLP}_1^l\left(\mathrm{MLP}_2^l(\mathbf{h}_i^l) + \sum_{j\in\mathcal{N}(i)}\mathbf{m}_{ij}^l\right),
\tag{5}
$$

where $\mathbf{h}_i^l$ denotes the $i$-th node-vector at the $l$-th layer, and $\mathbf{m}_{ij}^l$ denotes the message from the $j$-th node connected to the $i$-th node. The message is constructed using both atom-pair distances $d_{ij}$ and edge-features $\mathbf{f}_{ij}^{\mathrm{rxn}}$:

$$\mathbf{m}_{ij}^l = \mathrm{MLP}_2^l(\mathbf{h}_j^l)\left(\pi_{\mathrm{rbf}}^l(d_{ij}) \odot \pi_{\mathrm{edge}}(\mathbf{f}_{ij}^{\mathrm{rxn}})\right), \qquad (6)$$

where $\pi_{\mathrm{rbf}}^l$ and $\pi_{\mathrm{edge}}$ denote a radial basis kernel and an edge-feature embedding function, respectively, and $\odot$ denotes element-wise multiplication. A node-embedding function $\pi_{\mathrm{node}}$ is used to generate the initial node-vector $\mathbf{h}_i^1$ by processing a node-feature vector of the reaction graph $\mathbf{f}_i^{\mathrm{rxn}}$. The node-features and edge-features of the reaction graph are first constructed by concatenating those of $\mathcal{G}_R$ and $\mathcal{G}_P$, respectively: $\mathbf{f}_i^{\mathrm{rxn}} = \mathbf{f}_i^R \oplus \mathbf{f}_i^P$ and $\mathbf{f}_{ij}^{\mathrm{rxn}} = \mathbf{f}_{ij}^R \oplus \mathbf{f}_{ij}^P$. The edge-features contain bond-type information, and the node-features contain various atomic information, such as aromaticity, formal-charge, hybridization, valency, chirality, and whether the atom is in a ring. When an edge does not exist on one side of $\mathcal{G}_R$ or $\mathcal{G}_P$, the edge-feature of the molecular graph is adjusted with a zero feature vector. To utilize the reaction graph more informatively, we extended $\mathcal{G}_R$ and $\mathcal{G}_P$ to include edges within the 3-hop and encoded their edge-features based on the graph-distances. In addition, we added edges between nodes within cutoff radius $\tau$ of the pairwise distances and assigned zero vectors as edge-features. All hyperparameters of TSDiff are summarized in the Supplementary Notes.

In summary, the neural network approximates the score function $\varepsilon$ by utilizing the noisy geometry $\mathcal{C}_t$ and the reaction graph $\mathcal{G}_{\mathrm{rxn}}$, resulting in less noisy geometry $\mathcal{C}_{t-1}$. In practice, it first computes the score function on the distance coordinate using the outcomes from the last layer and the edge information. These changes are then converted to atomic position changes via the chain rule. The overall inference phase is a Markov process, iteratively sampling $\mathcal{C}_{t-1}$ from $p_\theta(\mathcal{C}_{t-1}|\mathcal{C}_t, \mathcal{G}_{\mathrm{rxn}})$, starting from noise $\mathcal{C}_T \sim \mathcal{N}(0, I)$ and resulting in $\mathcal{C}_0$. Further details of the sampling algorithm are described in the Supplementary Methods.

## Data

In this study, we used a publicly available chemical reaction dataset provided by Grambow et al.[43]. This consists of gas-phase elementary reactions involving up to seven C, O, or N atoms per molecule. Reactants were sampled from GDB-7, a subset of GDB-17[66]. The dataset contains a wide range of reactions with up to six bond changes, and most reactions occur with two or three bond changes, where the number of bond changes only counts the changes in connectivity between atoms, regardless of bond order. It also includes reactions with a wide range of barrier energies, up to 200 kcal mol$^{-1}$, to ensure that the model is not biased toward reactions with low energy barriers.

The reaction pathways were first elucidated using DFT calculations, specifically the single-ended GSM, and the TS geometries were computed using the saddle point optimization at the same level of theory. There are two types of datasets calculated by different DFT methods, namely B97-D3/def2-mSVP and $\omega$B97X-D3/def2-TZVP, and we used the $\omega$B97X-D3 dataset. Out of the 11,961 reactions in the $\omega$B97X-D3 dataset, we excluded two that involved non-reactive molecular nitrogen. Since our model only captures reaction information with 2D graphs, including these non-reactive molecules could lead to erroneous graph-embedding. We used a total of 11,959 reactions and randomly split the dataset in a ratio of 8:1:1. To improve the performance of our model, we also augmented our training data by including reverse reactions, i.e., swapping the reactants and products, resulting in a total of 19,132 training data points.

## Measurement details

To measure the accuracy of the generated TS geometries, we used the D−MAE metric, which is the MAE of the interatomic distances. The D−MAE between two different geometries, $\mathcal{C}$ and $\hat{\mathcal{C}}$, is defined as

$$\mathrm{D\text{-}MAE}(\mathcal{C}, \hat{\mathcal{C}}) = \frac{2}{N_{\mathrm{atom}}(N_{\mathrm{atom}} - 1)} \sum_{i<j}^{N_{\mathrm{atom}}} |d_{ij} - \hat{d}_{ij}|, \qquad (7)$$

where $d_{ij}$ and $\hat{d}_{ij}$ denote the interatomic distances between the $i$-th and $j$-th atoms of $\mathcal{C}$ and $\hat{\mathcal{C}}$, respectively, and $N_{\mathrm{atom}}$ is the number of atoms. We performed an atom index alignment between $\mathcal{C}$ and $\hat{\mathcal{C}}$ to minimize the D−MAE between them. This is necessary to match the indices of nodes that are indistinguishable on the molecular graph, such as hydrogen in a methyl group.

To evaluate TSDiff from the perspective of a generative model, we used the COV and MAT scores. The two scores are defined as

$$\mathrm{COV}(S_{\mathrm{gen}}, S_{\mathrm{ref}}) = \frac{1}{|S_{\mathrm{ref}}|} \left| \left\{ \mathcal{C} \in S_{\mathrm{ref}} | \, \mathrm{D\text{-}MAE}(\mathcal{C}, \hat{\mathcal{C}}) < \delta, \hat{\mathcal{C}} \in S_{\mathrm{gen}} \right\} \right|, \qquad (8)$$

$$\mathrm{MAT}(S_{\mathrm{gen}}, S_{\mathrm{ref}}) = \frac{1}{|S_{\mathrm{ref}}|} \sum_{\mathcal{C} \in S_{\mathrm{ref}}} \min_{\hat{\mathcal{C}} \in S_{\mathrm{gen}}} \mathrm{D\text{-}MAE}(\mathcal{C}, \hat{\mathcal{C}}), \qquad (9)$$

where $S_{\mathrm{gen}}$ and $S_{\mathrm{ref}}$ denote the sets of generated and reference geometries, respectively, and $\delta$ denotes a criterion value, and $|\cdot|$ means the number of elements in a given set. Note that the number of the reference geometry for each reaction is one, which means $|S_{\mathrm{ref}}| = 1$ in our evaluations.

The COV score is variable depending on the choice of $\delta$. We first adopted the value of 0.1 Å based on the accuracy of a state-of-the-art model. To validate this choice, we investigated the D−MAE distribution of the generated geometries with their optimized results as a reference. We confirmed that approximately 25% of the samples have a D−MAE greater than 0.1 Å, despite being well driven to their reference, suggesting that the criterion may be stringent. To mitigate this, we further evaluated the COV score with $\delta$ of 0.2 Å, which gives less than 4% of such cases.

## Computational details

To validate the TS conformations generated by TSDiff, we used two quantum chemical calculations based on DFT and assessed the success rates in the two calculations. All quantum chemical calculations in this study were performed with Orca[67] at the $\omega$B97X-D3/def2-TZVP level of theory, the same as in Grambow's database[43].

The saddle point optimization was performed using the Berny algorithm[20], and the detailed options are as follows. The maximum number of iterations was set to 200, and the Hessian was computed in the first optimization step only. The convergence criteria for the gradients, displacements, and energies were set to 3e-4, 4e-3, and 5e-6 in atomic units, respectively. To evaluate the success of the saddle point optimization, we verified that the computations converged and the resulting geometry had a single imaginary frequency lower than −100 cm$^{-1}$, which is the same criterion as in the case of Grambow et al.[43].

The maximum number of iterations for both the forward and backward IRCs was set to 200, and the convergence criterion for the gradients was set to 2e-3 in the atomic unit. Upon completion of the IRC, to ensure that the resulting reactant and product geometries successfully matched those of the reference, we checked the consistency of their molecular connectivity using Open Babel[68].

**Reporting summary**

Further information on research design is available in the Nature Portfolio Reporting Summary linked to this article.

## Data availability

The organic gas-phase reaction database used in this study is available at https://doi.org/10.1038/s41597-020-0460-4. The TS geometries generated in this study and their quantum chemical calculation results have been deposited in a Zenodo repository at 10.5281/zenodo.10224071[69]. Source data are provided with this paper.

## Code availability

An implementation of the proposed model, TSDiff, is available at https://github.com/seonghann/tsdiff[70].

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

## Acknowledgements

This work was supported by Korea Environmental Industry and Technology Institute (Grant No. RS202300219144) to S.K., J.W., and W.Y.K., the National Research Foundation of Korea funded by the Ministry of Science and ICT (Grant No. 2018R1A5A1025208) to W.Y.K., and Samsung Electronics Co., Ltd (Grant No. IO201208-07820-01) to S.K. and W.Y.K.

## Author contributions

S.K. and J.W. equally contributed to this work. They designed the methodology and performed quantum calculations. All authors wrote the manuscript together and W.Y.K. supervised the project.

## Competing interests

The authors declare no competing interests.
