## [Peer Review File · Nature Communications]

Diffusion-based Generative AI for Exploring Transition States from 2D Molecular GraphsREVIEWER COMMENTS

Reviewer #1 (Remarks to the Author):

The paper by Kim et al describes the initial development and testing of an AI model for transition state (TS) prediction for organic chemical reactions. The paper is well-written throughout, and generally well-presented too. The length is generally appropriate (although see below). I would anticipate that this research would be of interest to computational chemists - the difficulty of finding TSs for analysis of chemical reactions is well-known, and new ideas in this field are welcome.

The AI model generated here seems capable of predicting TSs with some reliability and structural accuracy, as indicated in the results presented, although I have some question about the wider applicability of this approach below. Overall, the results presented seem to support the broad conclusions drawn, although some further points of comparison with existing methods would also be welcome, as noted below.

The methodology appears sound, although I did have some difficulty in following what it actually is that the authors are doing when generating TS structures - some comments below touch on this. The overall idea, using AI to predict TS structures, seems to be an evolution of recent work in this area, as the authors note - the key feature here is that the authors approach can sample conformations of TS structures.

Further comments:

(1) The obvious drawback of the authors approach is that it requires training data. They have used the well-known Grambow dataset - but that is limited to "organic" chemistry, and not particularly representative of a wider class of possible reactions. So - although the authors suggest their model is computationally efficient - they have not factored in the enormous cost of generating training data. Some further comment on this important point would be welcome - for example, how do they intend to generate this data for "non-organic" reactions? (e.g. homogeneous catalysis by organometallic complexes).

(2) The authors should read through and make sure that vague sentences are removed or corrected. For example, in the Introduction, it notes that "TSDiff achieved significantly high success rates of 97.4% and 90.6%..." - but success in what?

(3) The description of the methodology is a little unclear to me - and I expect it would be quite unfathomable to many who are not experts in current AI. For example, what is the "denoising network" actually doing in practice? Is it simply sampling conformations, subject to obeying a target reaction graph?

(4) As a general comment, it is not clear to me what the "use case" of this approach is. Do practitioners in studying reactive chemistry really want to be presented with hundreds of possible conformers for a given TS? What would they be expected to do with these? Would further TS refinement (computationally expensive!) be required for each in order to really be certain that an appropriate TS is found? Some further insight into how this approach will work in an automated reaction chemistry analysis workflow would be very welcome.

(5) Figure 4 raises a question about how TSDiff works. For the given reaction, it appears that this is essentially a 1-D problem - the C-C distance appears to serve as a good reaction coordinate, and the structures generated by TSDiff really just differ in the conformation of a distal O-H functional group. How representative of the Grambow dataset is this? In other words, is TSDiff learning a low-dimensional representation (1-D or 2-D?) of a TS given reactants and products, then building a conformational probability distribution on top of that? Figure 4 makes me think this is the case - if so, that's fine, but it would be useful to have some further analysis for more "involved" reactions (i.e. reactions where there isn't a "simple" reaction coordinate).

(6) In Table 1, at 10 samplings, nearly 20% of the TSs in the data set are "missed" - at 100 samplings this decreases to 10% missed. This seems like a lot (i.e. 100s of TSs missed) - some comment is needed here, especially if this approach would be used as part of a practical workflow. For example, how would one know when to stop sampling?

(7) In Table 2, why isn't TSDiff used to calculate D-MAE?

(8) There are a large number of schemes that have been proposed to generate "good" TS structures by interpolation between reactants and products - things like IDPP, and Todd Martinez's geodesic interpolation, for example. Furthermore, these methods are generally "chemistry agnostic", in that they don't need a training set like TSDiff does. Some comparison of how TSDiff compares to the TS structures generated by these schemes would be useful.

So - overall - this is an interesting paper, although I have a few comments about validation and wider applicability. This does not seem to be something that others could generally use "off the shelf" without some further development - but is interesting nonetheless.

Reviewer #2 (Remarks to the Author):

MS: 429121_0_art_file_7617039_rpffv8_5

Title: Diffusion-based Generative AI for Exploring Transition States from 2D Molecular Graphs

Authors: Seonghwan Kim, Jeheon Woo and Woo Youn Kim

In this work, the authors propose a stochastic generative model to investigate transition state (TS) conformations using only 2D graphs as inputs. The model is trained on a dataset consisting of 11,959 reactions provided by Grambow et al. The results, as evaluated via appropriate matrices such as D-MAE, COV, and MAT, demonstrate the high-quality performance of the model and validate its ability to generate accurate TS conformations. Additionally, the predicted TS conformations are further validated through the support saddle point optimizations and IRC of IRC paths.

Whereas minimum searches on potential energy surfaces (PES) have become more or less a routine task, this does not hold for TS searches. Often, finding the correct TS connecting reactants and products under consideration strongly depends on the topology of the underlying PES (which can be

complex), the sophistication of the search algorithm, a good guess to start from as well as additional input such as information about the stoichiometry of the TS from kinetic data or details about the atoms involved in the bond-forming/breaking processes, and last but not least on good chemical intuition.

This work provides a promising novel effective tool to identify TS conformations without relying on expensive quantum chemical calculation and/or time-consuming trial and error procedures the theoretical chemistry community is looking for. Therefore, this referee recommends publication of this manuscript after considering the following points:

1. It would be beneficial to include information about the specific types of reactions in the dataset used for this study, such as hydrogen transfer reactions, hydrogen addition reactions, multiple bond reactions, and transition metal-related reactions., etc. This would provide insights into the composition of the dataset and the range of reactions considered.
2. To enhance the understanding of the model's applicability, it would be valuable to discuss the types of TS conformations that can be effectively calculated using the proposed model.
3. The manuscript lacks detailed information about the architecture and fine-tuning parameters of the TSDiff model. Adding more comprehensive explanations about the model's architecture and the specific parameters used for fine-tuning could considerably increase the value of this work
4. In the Data section, "GDB-7" should be corrected to "GDB-17" to accurately refer to the dataset used in the study.

Reviewer #3 (Remarks to the Author):

Characterizing transition states is one of the key objectives of computational chemistry due to their utility in determining reaction kinetics. Conventionally, human intuition coupled with expensive quantum chemical potentials have been used to optimize transition state geometries in an inherently manual process. Sophisticated automated transition state finding methods are becoming increasingly common, and as availability of computationally generated data increases, newly developed methods often employ machine learning techniques. In this work, the authors develop the first diffusion model for TS generation requiring only 2D graphs of reactants and products. They demonstrate that geometries can be generated accurately and with high success rates. Overall, this work is an important contribution to the field, and I recommend publication subject to additional discussion and evaluation.

While the authors are relatively thorough in their motivation and discussion of their method, some major topics deserve further attention:

- 1) Why is a generative method necessary for TS geometry prediction? Many TSs have very well-defined reaction coordinates that proceed through a single distinct TS (at least in the subspace

corresponding to the bonds changing during the reaction). Prior discriminative methods capitalize on this fact by prioritizing the coordinates that correspond to bond changes. Would such inductive bias also be useful for a generative model? The authors highlight the ability to sample distinct TS geometries, and it is indeed a very interesting result that conformational diversity is possible given the single TS geometries in the training data. While reactions with multiple TSs certainly exist, i.e., where distinct yet energetically similar reaction coordinates are possible, there is no evidence that such reactions can be sampled by the authors' method. For example, Fig. 3 and 4 show several TS geometries, yet all of them only involve conformational changes of the coordinates not involved in the bond changes. It is therefore likely that most of the lower-energy TSs relative to the reference reaction correspond to different reactant and product geometries and are not actually lower-lying TSs of these reactions. Additional discussion on this topic is warranted.

2) The evaluation metrics and comparison to existing methods deserve additional attention. D-MAE considers all atoms in the molecules, but it would also be interesting to compute a version of the metric that only considers atoms in the reactive center to assess performance in the most critical part of the TS. Additionally, why was D-MAE chosen compared to standard metrics like RMSD? Would it make sense to include RMSD-based metrics? As already noted by the authors, COV and MAT only have limited application when the reference ensemble only has a single member. Clearly, increasing the number of samples (Table 1 and Fig. 5) leads to arbitrarily good results. In a real scenario where reference data is not available, which of the samples should be selected for further consideration? Are most of the generated samples valid transition states (that just differ in terms of the non-reactive coordinates, see point 1)? Additional discussion and evaluation is required before being able to make the claim that the authors' method outperforms all others. For example, it would make sense to also compute precision versions of COV and MAT (where reference and generated ensembles are switched) to assess how many of the generated TSs are of high quality.

3) Additional explanation is required in the methods section. How is the position score function converted to a pairwise distance score function? The authors use the term "distance", which generally implies a scalar value, but in some places, it seems like this term refers to the vector between the positions of two atoms, i.e., the distance vector, is that correct? How does converting changes in node-pair distances (or distance vectors) to positions work considering that there are N^2 distances but only $N \cdot 3$ Cartesian coordinates?

Minor points:

- One motivation mentioned by the authors for using 2D graphs as input instead of 3D coordinates is computational cost. However, cost-efficient and highly-performant ML models for equilibrium geometry prediction exist. If such approaches are used, cost is less of a concern and the additional information provided by 3D structures could be utilized. What are the authors' thoughts on this? If the concern is related to sensitivity to specific input structure instead, could a method be developed that uses the reactant/product ensembles instead? Alternatively, this sensitivity to input structure could be a desirable feature in order to obtain a TS geometry with the same conformation in the parts of the molecules that are not involved in the reactive center.

- Can the percentage of correctly generated TSs be included in Table 2? It would be interesting to compare this metric across the different methods.

- TS-GAN (Makos et al.) is also a generative method. Is it possible to initialize it with different random

vectors to sample multiple TSs? If so, a better comparison to the authors' method would be possible.

- Some discussion on computational cost would be useful. The authors' method follows GeoDiff to use 5000 diffusion steps, which is expensive. How does the computational cost compare to the other methods?

- The wording in the first sentence in the introduction is unusual; I suggest revising this.

Response to Review 1.

We thank the reviewer for the careful evaluation and valuable comments that have helped us improve the quality of our manuscript. Our point-by-point responses are attached below, and the manuscript has been revised accordingly.

Detailed comments

1. The obvious drawback of the authors approach is that it requires training data. They have used the well-known Grambow dataset - but that is limited to “organic” chemistry, and not particularly representative of a wider class of possible reactions. So - although the authors suggest their model is computationally efficient - they have not factored in the enormous cost of generating training data. Some further comment on this important point would be welcome - for example, how do they intend to generate this data for “non-organic” reactions? (e.g. homogeneous catalysis by organometallic complexes).

Reply:

We appreciate your comment about the limitations of our approach, which relies on training data. You rightly pointed out the Grambow dataset is confined to organic chemistry and may not adequately represent a wider range of reactions. We fully acknowledge this limitation and recognize the need for further research to extend our approach to non-organic reactions, such as homogeneous catalysis by organometallic complexes. In response to your comment, we have included a discussion about the limitations in the revised version of the manuscript. We have also introduced non-organic reaction datasets that are currently available, highlighting their potential to contribute to the expansion of training data and improve the performance and utility of machine learning approaches.

Revision:

4th paragraph in page 15.

However, it is important to recognize a limitation of this work, particularly its current restriction to organic reactions. Although inorganic databases exist, such as the FH51 set in GMTKN55, which contains 51 reactions in small inorganic and organic systems, and another database containing about 400 reactions including transition metals, the lack of large inorganic reaction databases limits the applicability of machine learning approaches in this domain. Nevertheless, with the ongoing accumulation of data in the future, we anticipate that the utility of TSDiff will expand to encompass a broader range of chemical reactions, including those involving inorganic species.

2. The authors should read through and make sure that vague sentences are removed or corrected. For example, in the Introduction, it notes that “TSDiff achieved significantly high success rates of 97.4% and 90.6%...” - but success in what?

Reply:

Regarding the reviewer’s concern about vague sentences, we apologize for any confusion. The success rates in that sentence refer to the ratio of generated samples that are validated as successful in the two calculations, the saddle point optimization and the IRC. Specifically, the success of the saddle point optimization refers to generated samples being successfully optimized to the respective saddle points with a single imaginary frequency, and the success

of IRC refers to the successfully optimized saddle points being validated by the IRC calculation, meaning that the molecular connectivities of the resulting compounds at the forward and backward points of IRC match those of the input. More detailed options, such as convergence criteria, are provided in the “Calculation details” section. The sentence the reviewer mentioned has been edited for clarity. We have also improved the “Computational details” section for clarity.

Revision:

4th paragraph in page 3.

The validity of the multiple TS conformations generated by TSDiff was verified by quantum chemical calculations based on DFT. First, saddle point optimization was performed on the generated geometries to obtain the TS geometries with a single imaginary vibrational frequency. The intrinsic reaction coordinate (IRC) calculation was performed to validate that the TS geometries correspond to the given graphically defined reaction. A detailed validation methodology is provided in the “Computational details” section. TSDiff achieved a significantly high success rate of 90.6 % in this validation, showing its reliability as an initial TS geometry guesser.

3. The description of the methodology is a little unclear to me - and I expect it would be quite unfathomable to many who are not experts in current AI. For example, what is the “denoising network” actually doing in practice? Is it simply sampling conformations, subject to obeying a target reaction graph?

Reply:

Thanks for pointing this out. We have recognized the lack of explanation regarding the denoising network in the previous text and have made the necessary improvements to enhance the clarity of the methodology.

The denoising network, as shown in Figure 1, denoises the noisy input geometry based on the graph information of the chemical reaction. In the inference step, the denoising network is applied throughout the inverse of the diffusion process (i.e., the denoising process). Here, the starting geometry is randomly initialized, and the geometry is progressively refined by the denoising network throughout the denoising process. After a predefined denoising step (5000 was used in this work), the predicted TS conformation is obtained.

To provide a clearer explanation, we have revised “A brief description of the generation process” and “Methods” sections and added Supplementary Information.

Revision:

1st paragraph in page 5.

TSDiff is based on the stochastic denoising diffusion method, where the model is trained to learn the reverse process of a noise process that adds a random noise to the given geometry at each discrete time step. At the inference phase, TS geometries are generated from an initial state with a complete noise through the iterative denoising process, where the noisy input is gradually refined by the denoising neural network at each time step, given the 2D reaction information (see Fig. 1b).

3rd paragraph in page 5.

The construction of a geometric reaction graph involves adding the noisy positions to the 2D reaction graph and connecting nodes with interatomic distances smaller than a specified cutoff radius. This process integrates bond information, graph distance information, and spatial

distance information as edge-features in the geometric graph. Subsequently, the model leverages these geometric reaction graphs to approximate a score function, a gradient of log-likelihood for noisy TS conformations, which is applied to denoising by updating the noisy positions towards the correct TS geometry.

4. As a general comment, it is not clear to me what the “use case” of this approach is. Do practitioners in studying reactive chemistry really want to be presented with hundreds of possible conformers for a given TS? What would they be expected to do with these? Would further TS refinement (computationally expensive!) be required for each in order to really be certain that an appropriate TS is found? Some further insight into how this approach will work in an automated reaction chemistry analysis workflow would be very welcome.

Reply:

As an example of the practical use case of TSDiff, let us consider the task of finding the TS of a given reaction. When using a typical double-ended TS search method, users first need to obtain the conformational geometries of reactants and products and their proper alignments. Then, starting from each reaction conformation out of them, a TS can be explored using the double-ended TS method. Finally, an energetic comparison can be made to determine the preferred reaction pathway. This process is very cumbersome because one has to consider many possible conformations and their appropriate alignments. TSDiff can greatly simplify this process. The user can obtain approximated TS conformations without the laboratory process.

As illustrated in Figure 4, generated TS samples tend to cluster together with similar conformations. Here, we used the t-SNE based on the interatomic distances of the TS geometries to plot the figure. This clustering tendency allows users to perform efficient computations without having to perform quantum calculations on each individual sample. This clustering can be accomplished in an automated fashion, for example using affinity propagation or Ward’s clustering methods. By selecting representative samples from each cluster, the users can significantly reduce the computational cost while still capturing various TS conformations.

While quantum calculational refinement of a few sampled conformations will still be required, we emphasize the need for TS conformation search in this study. The most favored reaction pathway can be found by validating candidate TS conformations.

In addition, it is worth noting that TS conformations generated by TSDiff have shown high success rates in subsequent TS optimizations. This indicates that TSDiff can effectively alleviate the time-consuming trial-and-error procedures of TS exploration.

We have revised the manuscript to take these points into account.

Revision:

4th paragraph in page 3.

Based on these results, we expect that TSDiff can greatly alleviate the time-consuming trial-and-error procedures of TS exploration.

1st paragraph in page 7.

On the t-SNE projection map, it is evident that similar conformations tend to cluster together and be closely located to their respective optimized results. This character of generated samples suggests that an efficient search for TS conformations is possible without having to perform quantum chemical calculations on the entire generated samples. Many clustering algorithms are already available, offering an effective means to select representative

conformation samples.

3rd paragraph in page 15.

One of the main advantages of TSDiff is its ability to find TSs without considering the conformations of the reactants and products and their alignments. Since TSDiff does not rely on specific conformations, it allows efficient exploration of TSs in graphically defined reactions with a more generalized approach. We demonstrated its usefulness in chemical reaction analysis by generating diverse, high-quality TSs based on a given molecular connectivity.

5. Figure 4 raises a question about how TSDiff works. For the given reaction, it appears that this is essentially a 1-D problem - the C-C distance appears to serve as a good reaction coordinate, and the structures generated by TSDiff really just differ in the conformation of a distal O-H functional group. How representative of the Grambow dataset is this? In other words, is TSDiff learning a low-dimensional representation (1-D or 2-D?) of a TS given reactants and products, then building a conformational probability distribution on top of that? Figure 4 makes me think this is the case - if so, that's fine, but it would be useful to have some further analysis for more "involved" reactions (i.e. reactions where there isn't a "simple" reaction coordinate).

Reply:

TSDiff, a diffusion model, is trained to restore a reference TS geometry (C_0) from a completely randomly distributed geometry (C_r) based on the given graph information. Therefore, it is not the case that TSDiff builds conformational probabilities after learning a low-dimensional representation.

The limited examples (such as Figure 4) in the previous text might lead the reader to believing that TSDiff generates only trivial or simple conformations. TSDiff can also generate TS conformations with different reactive coordinates. We have added a new section and figure to show these examples with more complex reaction coordinates (see Figure 7 in the revised manuscript).

Revision:

Section 2.6 "Analysis on multiple reaction pathways explored by TSDiff" in page 13-15.

We present additional results for four reactions in which TSDiff discovered TS conformations with distinct reaction coordinates. For each reaction, two TS conformations were obtained by saddle point optimization of the generated TSs, resulting in D-MAE values between two conformations of the generated TSs, resulting in D-MAE values between two conformations of 0.29 Å, 0.30 Å, 0.26 Å, and 0.15 Å for the four reactions, respectively.

...(omission of further details).

6. In Table 1, at 10 samplings, nearly 20% of the TSs in the data set are "missed" - at 100 samplings this decreases to 10% missed. This seems like a lot (i.e. 100s of TSs missed) - some comment is needed here, especially if this approach would be used as part of a practical workflow. For example, how would one know when to stop sampling?

Reply:

The COV score is a variable value depending on its criterion value. The value of 0.1 Å that we used corresponds to the highest accuracy of the existing models, so it might be a strict

criterion. Indeed, in our evaluation based on quantum chemical calculations, we observed a significant number of samples with D-MAE greater than 0.1 Å among the samples that succeeded in TS optimization. We have included a discussion of this point in the manuscript and added COV values to Table 1 using a less strict criterion of 0.2 Å. These results confirm that a larger fraction is covered by TSDiff.

Revision:

2nd paragraph in page 8.

The COV score measures the percentage of reference TS geometries covered by the predicted ones by TSDiff, where a reference is considered to be covered if there exists any predicted one having a D-MAE within a criterion of δ with the reference. We used two criterion values: $\delta = 0.1 \text{ \AA}$ and $\delta = 0.2 \text{ \AA}$.

3rd paragraph in page 19.

The COV score is variable depending on the choice of δ . We first adopted the value of 0.1 Å based on the accuracy of a state-of-the-art model. To validate this choice, we investigated the D-MAE distribution of the generated geometries with their optimized results as a reference. We confirmed that approximately 25% of the samples have a D-MAE greater than 0.1 Å, despite being well driven to their reference, suggesting that the criterion may be stringent. To mitigate this, we further evaluated the COV score with δ of 0.2 Å, which gives less than 4% of such cases.

7. In Table 2, why isn't TSDiff used to calculate D-MAE?

Reply:

Unlike previous works that predict a particular TS geometry from the given reactant and product geometries, TSDiff generates a range of TS conformations that can be formed from the connectivity information of the reactant and product. Since there is only one TS conformation per dataset used, there is no corresponding reference geometry for each TS conformation by TSDiff. For this reason, TSDiff was evaluated using COV and MAT, a widely used metric for evaluating generation methods.

To use D-MAE for the direct comparison, we identified samples among generated TS conformations that match the reference TS conformation on a subset of the test set, which have been determined by saddle point optimizations on the generated samples. Consequently, we calculated the D-MAE of the matched samples which was a subset of the test data: 53.2% and 84.6% covering rates of the test reactions with one and eight sampling rounds, respectively. The resulting D-MAE values were significantly low at 0.063 Å and 0.067 Å, respectively, indicating the high accuracy of TSDiff, which surpasses the performance of existing models. In the case of the single round sampling without the conformer matching (100% coverage), the D-MAE value was 0.137 Å, which is still lower than those of all models except Choi's one.

As a result, we have included the D-MAE values in Table 2 so that TSDiff can be compared with other models using the same metric. We have also included a detailed description of this in the manuscript.

Revision:

3rd paragraph in page 10 and 2nd paragraph in page 11.

Before comparing the accuracy of the models, it is important to note that evaluating the accuracy of TSDiff under the same conditions as existing models is not straightforward because TSDiff can generate other TS conformations, while we only have a single TS conformation

available as a reference. To address this, we evaluated the accuracy of TSDiff on a TS conformation only if it directly matches the corresponding reference. To identify these matching samples, we conducted saddle point optimizations on the generated samples. For a reliable comparison, we aimed to find as many test reactions with a matched reference as possible, so we performed the optimization on TS conformations generated with a total of eight rounds of sampling for each reaction. Samples with a D-MAE between their optimized result and the reference of less than 0.01 Å were considered matching, resulting in the covering rates of 53.2% and 84.6% of the test reactions with one and eight sampling rounds, respectively. One of the generated samples is randomly selected to calculate the D-MAE when multiple samples match the same reference TS in a given reaction graph, which gives a consistent D-MAE value regardless of the number of sampling rounds to facilitate fair evaluation.

Table 2 shows the D-MAE values of TSDiff with and without considering conformer matching. The latter is the case for a single sampling, the resulting conformation of which can be considered an approximate TS for the respective reference. In this case, the D-MAE value is 0.137 Å, which is lower than those of all models except Choi's one, indicating that TSDiff is fairly accurate when only providing a single TS without conformer matching. Furthermore, considering the conformer matching, the D-MAE values become 0.063 Å and 0.067 Å for one and eight sampling rounds, respectively, which are considerably lower than those of all. Note that while the covering rate increased from 53.2% to 84.6%, the D-MAE value remained consistent, suggesting its reliability as a metric to assess accuracy. Thus, it can be concluded that TSDiff generates TS geometries with better accuracy than the existing models, without computationally expensive 3D geometric information. Further analysis and methodology involving quantum calculations will be presented in subsequent sections.

8. There are a large number of schemes that have been proposed to generate “good” TS structures by interpolation between reactants and products - things like IDPP, and Todd Martinez's geodesic interpolation, for example. Furthermore, these methods are generally “chemistry agnostic”, in that they don't need a training set like TSDiff does. Some comparison of how TSDiff compares to the TS structures generated by these schemes would be useful.

Reply:

Before comparing TSDiff with other methods, it is important to highlight a unique feature of TSDiff. Traditional methods rely on the predetermined reactant and product geometries (also for previous machine learning models and interpolation-based methods). They end up with different TSs if the search begins with different conformations of reactants and products and their relative orientations. In contrast, TSDiff is a generative model that generates various TS conformations based on just molecular connectivity information. Therefore, for a fair comparison, one may need to first sample various reaction conformations for each given reaction, which is not easy for a large amount of reactions. Although we included the comparison with existing machine learning approaches, our focus lies more on the generation of diverse TS conformations rather than just predicting a single reference TS geometry. We hope the reviewer appreciates that we are maintaining the current evaluation approach.

Response to Review 2

We thank the reviewer for the careful evaluation and valuable comments that have helped us improve the quality of our manuscript. Our point-by-point responses are attached below, and the manuscript has been revised accordingly.

Detailed comments

1. It would be beneficial to include information about the specific types of reactions in the dataset used for this study, such as hydrogen transfer reactions, hydrogen addition reactions, multiple bond reactions, and transition metal-related reactions., etc. This would provide insights into the composition of the dataset and the range of reactions considered.

Reply:

Grambow's dataset was not specifically constructed to focus on particular reaction types. Instead, it contains a diverse set of gas-phase organic reactions, encompassing various reaction types. The reactants used as inputs for the single-ended method were sampled from GDB-7. This selection allowed us to cover reactions involving possible bond changes among atoms of C, H, O, and N.

In the revised version of our paper, we have included the above information in the introduction section to provide clarity on the composition of the dataset and the diversity of reactions considered. Additionally, we have included more detailed information about the dataset in the Data section of the paper, providing insights into the composition and range of reactions encompassed in our study.

Revision:

4th paragraph in page 3.

In this study, the performance of TSDiff was evaluated using Grambow's dataset, a set of diverse gas-phase organic reactions generated with the single-ended GSM, where reactant molecules were sampled from GDB-7 to cover reactions involving possible bond changes among C, H, O, and N atoms.

2nd paragraph in page 18.

This consists of gas-phase elementary reactions involving up to seven C, O, or N atoms per molecule. Reactants were sampled from GDB-7, a subset of GDB-17. The dataset contains a wide range of reactions with up to six bond changes, and most reactions occur with two or three bond changes, where the number of bond changes only counts the changes in connectivity between atoms, regardless of bond order. It also includes reactions with a wide range of barrier energies, up to 200 kcal/mol, to ensure that the model is not biased toward reactions with low energy barriers.

2. To enhance the understanding of the model's applicability, it would be valuable to discuss the types of TS conformations that can be effectively calculated using the proposed model.

Reply:

TS conformations resulting from rotatable bonds in reactive coordinates are the most frequently found conformations. However, TSDiff can also generate other TS conformations that differ in non-reactive coordinates. The limited reaction examples in the previous

manuscript might lead the reader to believing that TSDiff only generates trivial or simple conformations. Therefore, we have included a new figure to show more examples with complex reaction coordinates (see Figure 7 in the revised manuscript). We have also added a comment on this in the Discussion section.

Revision:

3rd paragraph in page 15.

We demonstrated its usefulness in chemical reaction analysis by generating diverse, high-quality TSs based on a given molecular connectivity. This is amazing considering that TSDiff learned only one TS conformation for each reaction during the training phase. TSDiff has been able to effectively capture TS conformations resulting from rotatable bonds in non-reactive coordinates and different reaction coordinates.

3. The manuscript lacks detailed information about the architecture and fine-tuning parameters of the TSDiff model. Adding more comprehensive explanations about the model's architecture and the specific parameters used for fine-tuning could considerably increase the value of this work.

Reply:

We appreciate the valuable feedback regarding the lack of detailed information on the architecture and fine-tuning parameters of the TSDiff model. We have included a comprehensive table in the manuscript describing the detailed hyperparameters of the model's architecture (see Supplementary Table S1). We have also made the code used in our experiments openly available on GitHub. By sharing the code, we aim to provide the research community with a resource that further elucidates the implementation details of our model.

Revision:

2nd paragraph in page 20.

An implementation of the proposed model, TSDiff, is available at <https://github.com/seonghann/tsdiff>.

2nd paragraph in page 17.

Additionally, we added edges between nodes within cutoff radius τ of the pairwise distances and assigned zero vectors as edge-features. All hyperparameters of TSDiff are summarized in "Hyperparameters" section of the Supplementary Information.

1st paragraph in page 4 (Supplementary Information).

We here append all hyperparameters related to TSDiff including training hyperparameters (see Table S1).

4. In the Data section, "GDB-7" should be corrected to "GDB-17" to accurately refer to the dataset used in the study.

Reply:

We meant to say GDB-7, which is a subset of GDB-17. We modified the sentence for clarity.

Revision:

2nd paragraph in page 18.

This consists of gas-phase elementary reactions involving up to seven C, O, or N atoms per molecule. Reactants were sampled from GDB-7, a subset of GDB-17.

Response to Review 3.

We thank the reviewer for the careful evaluation and valuable comments that have helped us improve the quality of our manuscript. Our point-by-point responses are attached below, and the manuscript has been revised accordingly.

Detailed comments

1. Why is a generative method necessary for TS geometry prediction? Many TSs have very well-defined reaction coordinates that proceed through a single distinct TS (at least in the subspace corresponding to the bonds changing during the reaction). Prior discriminative methods capitalize on this fact by prioritizing the coordinates that correspond to bond changes. Would such inductive bias also be useful for a generative model? The authors highlight the ability to sample distinct TS geometries, and it is indeed a very interesting result that conformational diversity is possible given the single TS geometries in the training data. While reactions with multiple TSs certainly exist, i.e., where distinct yet energetically similar reaction coordinates are possible, there is no evidence that such reactions can be sampled by the authors' method. For example, Fig. 3 and 4 show several TS geometries, yet all of them only involve conformational changes of the coordinates not involved in the bond changes. It is therefore likely that most of the lower-energy TSs relative to the reference reaction correspond to different reactant and product geometries and are not actually lower-lying TSs of these reactions. Additional discussion on this topic is warranted.

Reply:

As the reviewer suggested, such inductive bias involving reaction coordinate information would be beneficial for reducing possible transition state conformations. Clearly, the concept of reaction coordinates in chemical reactions is common, particularly by focusing on bond changes. However, other coordinates also evolve during reactions. Unlike methods with a direct use of 3D input that implicitly contains a specific reaction coordinate, our model employs only a 2D graph representation in order to sample multiple potential reaction coordinates.

In the manuscript, we have demonstrated, through DFT calculations, the possibility of finding energetically favorable geometries. Clearly, the geometry of the reactants corresponding to the newly found TS may differ from the reference, and a lower energy TS does not necessarily indicate a lower activation barrier. However, discovering TSs with lower energy is still meaningful because it gives a TS corresponding to a lower overall barrier. Considering the fact that the barrier for conformational changes is typically lower than that for bond changes, we believe the reviewer will appreciate the current evaluation approach for barrier height.

Nevertheless, to address the concerns raised by the reviewer, we performed a comparison of the activation barriers between the reference reaction and newly sampled reactions, which entailed optimization of the reactants geometry in conjugation with saddle point optimization and IRC calculation. As a result, we observed that 513 among 998 reactions were more favorable compared to the reference. This represents a significant increase compared to the 309 reactions that had lower overall barrier heights than the reference. This means that the newly found reactants generally have higher energies than those of the reference reactants, which is mainly due to the conformational search for reference reactants performed by Grambow et al. We have included these discussions in the

manuscript.

We believe that sampling distinct TSs, even in coordinates other than bond changes, is valuable for conformational search purposes, thus justifying the necessity of a generative approach in TS search. However, it would be particularly fascinating to demonstrate the ability to sample distinct TS geometries specifically in the bond-change coordinates. Among the 998 successful cases in the IRC test, we have identified such cases, and as a result, we have included a section 2.6 of “Analysis on multiple reaction pathways explored by TSDiff” showcasing these TS geometries.

Revision:

2nd paragraph in page 12.

In our investigation of the energetics of the successfully validated TSs through IRC, we discovered 309 new TSs with energies lower than those of the reference TSs by more than 0.1 kcal/mol. ... (omission of further details). These findings highlight the fact that lower equilibrium geometries do not always correspond to reaction pathways with the lowest overall TS barriers.

Section 2.6 “Analysis on multiple reaction pathways explored by TSDiff” in page 13-15.

We present additional results for four reactions in which TSDiff discovered TS conformations with distinct reaction coordinates. For each reaction, two TS conformations were obtained by saddle point optimization of the generated TSs, resulting in D-MAE values between two conformations of the generated TSs, resulting in D-MAE values between two conformations of 0.29 Å, 0.30 Å, 0.26 Å, and 0.15 Å for the four reactions, respectively. ... (omission of further details).

2. The evaluation metrics and comparison to existing methods deserve additional attention. D-MAE considers all atoms in the molecules, but it would also be interesting to compute a version of the metric that only considers atoms in the reactive center to assess performance in the most critical part of the TS. Additionally, why was D-MAE chosen compared to standard metrics like RMSD? Would it make sense to include RMSD-based metrics? As already noted by the authors, COV and MAT only have limited application when the reference ensemble only has a single member. Clearly, increasing the number of samples (Table 1 and Fig. 5) leads to arbitrarily good results. In a real scenario where reference data is not available, which of the samples should be selected for further consideration? Are most of the generated samples valid transition states (that just differ in terms of the non-reactive coordinates, see point 1)? Additional discussion and evaluation is required before being able to make the claim that the authors' method outperforms all others. For example, it would make sense to also compute precision versions of COV and MAT (where reference and generated ensembles are switched) to assess how many of the generated TSs are of high quality.

Reply:

We selected D-MAE to measure geometric differences for comparison with existing methods. Bond distances in molecular geometry serve as important indicators, and we consider D-MAE to be more meaningful than measuring error in atomic positions. Nonetheless, we acknowledge the reviewer's suggestion of considering additional metrics. For instance, evaluating D-MAE specifically in the bond-change coordinate could provide insights into the generative performance in the reactive core region of the transition state. While we did

explore MAT values gauged by the core-region D-MAE as suggested, the results mirrored the existing MAT trend, with MAT values diminishing as sampling numbers increased. This suggests potential variations in the core-region conformations, and aligns with our findings (reply 1) of generated conformations that differ from each other in the bond-change coordinates. We hope the reviewer appreciates that our decision to illustrate an example chemical reaction (Figure 7), rather than introduce additional evaluation metrics, was made in an effort to clearly convey this generative performance of TSDiff.

Additionally, the question of selecting the number of samples or determining which samples to consider is subjective and depends on user's discretion. However, as observed in Figure 4, the geometries generated by TSDiff seem to cluster based on their conformations. This suggests that a clustering approach might be viable for sample selection. By clustering the generated geometries, we can select representative samples, which can serve as a conformational search strategy to minimize the need for extensive TS optimization and subsequent DFT calculations. We have included a discussion of this to the revised manuscript.

Regarding the evaluation method, given that we have only one reference TS per reaction, assessing precision is indeed challenging. In this context, interchanging the positions of generated and reference data may not effectively represent precision. Instead, we utilized quantum validation to assess the precision of the TSDiff model, with the IRC success rate in Table 3 representing coverage from a precision standpoint. We have included explanations to clarify the meaning of these values in our manuscript.

We recognize that comparing TSDiff to other methods requires careful consideration. As you rightly point out, MAT is well suited for checking recall, but is not directly comparable to the D-MAE metric, which measures the accuracy of predictive models. To address this, we have evaluated TSDiff using the D-MAE metric and performed comparisons with existing models based on this metric. Since TSDiff generates a wide range of TS conformations, for a fair evaluation, we considered the conformations of the generated TSs that match those of the corresponding reference TS, which was identified by saddle point optimization results. Specifically, we measured the D-MAE of TSDiff with and without considering the conformer matching.

In the case of D-MAE evaluation without considering conformer matching, a generated TS is considered as an approximate TS for the corresponding reference, resulting in a D-MAE of 0.137 Å. This value was still lower than all models except Choi's, indicating that TSDiff is fairly accurate even without conformer matching. Additionally, by performing one and eight sampling rounds in conjugation with conformer matching, TSDiff achieved the D-MAEs of 0.063 Å and 0.067 Å, respectively, with 53.2% and 84.6% of the test set being matched. These values were significantly lower than those of other models, further demonstrating the superior accuracy of TSDiff. Furthermore, the consistent D-MAE value despite the increased coverage from 53.2% to 84.6% underscores the reliability of the metric in assessing accuracy. Based on these findings, we conclude that TSDiff outperforms existing models in terms of accuracy. We have integrated this discussion into the revised manuscript.

Revision:

1st paragraph in page 7.

On the t-SNE projection map, it is evident that similar conformations tend to cluster together and be closely located to their respective optimized results. This character of the generated samples suggests that an efficient search for TS conformations is possible without having to perform quantum chemical calculations on the entire generated samples. Many clustering algorithms are already available, offering an effective means to select representative

conformation samples.

4th paragraph in page 11.

The IRC validation was carried out to verify that the optimized TS geometries are on the reaction pathways connecting the correct reactants and products. This verification process enables the validation of the precision of TSDiff. ... (omission of further details). For 998 reactions, corresponding to 83.4% of the test set, the IRC calculations successfully converged by linking the saddle point to the correct reactants and products, demonstrating the high precision of the TSDiff model.

3rd paragraph in page 10 and 2nd paragraph in page 11.

Before comparing the accuracy of the models, it is important to note that evaluating the accuracy of TSDiff under the same conditions as existing models is not straightforward because TSDiff can generate other TS conformations, while we only have a single TS conformation available as a reference. To address this, we evaluated the accuracy of TSDiff on a TS conformation only if it directly matches the corresponding reference. To identify these matching samples, we conducted saddle point optimizations on the generated samples. For a reliable comparison, we aimed to find as many test reactions with a matched reference as possible, so we performed the optimization on TS conformations generated with a total of eight rounds of sampling for each reaction. Samples with a D-MAE between their optimized result and the reference of less than 0.01 Å were considered matching, resulting in the covering rates of 53.2% and 84.6% of the test reactions with one and eight sampling rounds, respectively. One of the generated samples is randomly selected to calculate the D-MAE when multiple samples match the same reference TS in a given reaction graph, which gives a consistent D-MAE value regardless of the number of sampling rounds to facilitate fair evaluation.

Table 2 shows the D-MAE values of TSDiff with and without considering conformer matching. The latter is the case for a single sampling, the resulting conformation of which can be considered an approximate TS for the respective reference. In this case, the D-MAE value is 0.137 Å, which is lower than those of all models except Choi's one, indicating that TSDiff is fairly accurate when only providing a single TS without conformer matching. Furthermore, considering the conformer matching, the D-MAE values become 0.063 Å and 0.067 Å for one and eight sampling rounds, respectively, which are considerably lower than those of all. Note that while the covering rate increased from 53.2% to 84.6%, the D-MAE value remained consistent, suggesting its reliability as a metric to assess accuracy. Thus, it can be concluded that TSDiff generates TS geometries with better accuracy than the existing models, without computationally expensive 3D geometric information. Further analysis and methodology involving quantum calculations will be presented in subsequent sections.

3. Additional explanation is required in the methods section. How is the position score function converted to a pairwise distance score function? The authors use the term "distance", which generally implies a scalar value, but in some places, it seems like this term refers to the vector between the positions of two atoms, i.e., the distance vector, is that correct? How does converting changes in node-pair distances (or distance vectors) to positions work considering that there are N^2 distances but only N^3 Cartesian coordinates?

Reply:

We acknowledge that our explanation was insufficient regarding this matter. The distances of all atom-pairs connected by an edge could be interpreted as a vector, denoted as

$d = \{d_{ij}\}_{(i,j) \in \varepsilon}$. It's worthy to note that the final data distribution $p_\theta(C_0)$ should be invariant to SE(3) transformations. To ensure this, each transition must follow an SE(3) equivariant distribution.

For this, we utilized distance-coordinate which is invariant under the transformations. By Shi et al. [1], the equivariance of $\nabla_{C_t} \log q(C_t|\cdot)$ could be addressed by decomposing it into

$\frac{\partial d_t}{\partial C_t} \nabla_{d_t} \log q(d_t|\cdot)$. We assume that $\nabla_{d_t} \log q(d_t|d_0) = \frac{d_t - \sqrt{\alpha_t} d_0}{1 - \alpha_t}$. This allows us to rewrite the

target score function as $\frac{\partial d_t}{\partial C_t} \frac{d_t - \sqrt{\alpha_t} d_0}{1 - \alpha_t}$, and here, $\frac{\partial d_t}{\partial C_t}$ can be easily calculated using the relationship between the positions and the corresponding distance. Therefore, while the model technically predicts the score function in distance-coordinate, it is converted to score function in Cartesian-coordinate.

As you pointed out, there is redundancy in distance-coordinate, but this method has proven effective in our study, as it has in Xu et al. [2]. Additionally, it's worth considering approaches that use other internal coordinates, which would be our next research topic.

We have revised the manuscript and have included this discussion to clarify the diffusion method.

* references

[1]: Shi, C., Luo, S., Xu, M., & Tang, J. (2021, July). Learning gradient fields for molecular conformation generation. In International conference on machine learning (pp. 9558-9568). PMLR.

[2]: Xu, M., Yu, L., Song, Y., Shi, C., Ermon, S., & Tang, J. (2022). Geodiff: A geometric diffusion model for molecular conformation generation. arXiv preprint arXiv:2203.02923.

Revision:

3rd paragraph in page 16.

The distribution $p_\theta(C_0)$ should be SE(3) invariant. To ensure this, each backward transition is required to follow an SE(3) equivariant distribution, which can be addressed by utilizing the pairwise distances $d = \{d_{ij}\}_{(i,j) \in \varepsilon}$ (omission of further details).

4. One motivation mentioned by the authors for using 2D graphs as input instead of 3D coordinates is computational cost. However, cost-efficient and highly-performant ML models for equilibrium geometry prediction exist. If such approaches are used, cost is less of a concern and the additional information provided by 3D structures could be utilized. What are the authors' thoughts on this? If the concern is related to sensitivity to specific input structure instead, could a method be developed that uses the reactant/product ensembles instead? Alternatively, this sensitivity to input structure could be a desirable feature in order to obtain a TS geometry with the same conformation in the parts of the molecules that are not involved in the reactive center.

Reply:

Indeed, the advent of cost-effective and high-performance ML models is promising, and their application in generating equilibrium geometries seems beneficial. However, the 3D input

geometries need to be positioned on reaction coordinates to be ready for reactions, which are distinct from the typical equilibrium geometries. This introduces complexities, such as the need to specify the molecular conformations of the reactant/product pair simultaneously and their relative positioning in multi-molecular reactions. Considering an ensemble of inputs could be a viable approach, but the primary issue to address is ensuring that the reactant/product pair embodies a consistent reaction coordinate.

Meanwhile, if 2D graphs are employed to generate the input geometries, we believe there's minimal information difference compared to directly generating the TS from 2D graphs. If the dataset for the input geometry generation model were larger than the TS geometry data, it could potentially provide additional information. However, this is not the case with the data used in our study.

Nevertheless, if proper 3D coordinates could be obtained, they will certainly serve as good constraints for geometry prediction. However, from a generative perspective, we believe that 3D input geometry will limit generative diversity.

5. Can the percentage of correctly generated TSs be included in Table 2? It would be interesting to compare this metric across the different methods.

Reply:

Clearly, comparing the percentage of correctly generated transition states across different methods, as suggested by the reviewer, would be interesting. However, this poses a challenge as it necessitates costly DFT calculations to verify the correctness of the generated samples. Additionally, the datasets utilized in each study vary slightly, limiting the feasibility of borrowing the quantum evaluation values in the papers. We hope the reviewer appreciates our decision to maintain our current evaluation approach.

6. TS-GAN (Makos et al.) is also a generative method. Is it possible to initialize it with different random vectors to sample multiple TSs? If so, a better comparison to the authors' method would be possible.

Reply:

The primary objective of the TS-GAN model is to predict the specific transition state (TS) geometry corresponding to a given set of reactants and products. While TS-GAN utilizes a generative adversarial network (GAN) architecture, it is important to emphasize that its purpose is not to generate various TS conformations, but rather to accurately predict the reference TS geometry present in the dataset. In other words, it does not align with the reason why TSDiff was evaluated using generative metrics. In fact, in their original paper, TS-GAN was evaluated on a prediction task, not a generation task. This ensures that the evaluation is consistent with the intended purpose of the model as a predictive model.

7. Some discussion on computational cost would be useful. The authors' method follows GeoDiff to use 5000 diffusion steps, which is expensive. How does the computational cost compare to the other methods?

Reply:

Indeed, due to its iterative time steps, the diffusion model incurs a higher inferencing cost compared to other ML methods. With an ensemble of eight models, inference time for each chemical reaction was less than a few seconds, which is trivial compared to the cost of a DFT calculation. Furthermore, there has been active research, often referred to as "distillation," aimed at reducing diffusion steps with minimal loss of accuracy. We believe that this relatively high inferencing cost can be mitigated. We have added a discussion on the inferencing time of TSDiff.

Revision:

4th paragraph in page 5.

Diffusion models entail higher inference costs compared to other deep learning models, mainly due to their iterative denoising process. Specifically, TSDiff requires 5000 denoising steps for inference, which takes a few seconds per reaction. However, this cost is negligible compared to DFT calculations for TS optimization.

8. The wording in the first sentence in the introduction is unusual; I suggest revising this.

Reply:

We have revised that sentence.

Revision:

1st paragraph in page 2.

A transition state (TS) refers to a transient molecular configuration that places on top of the energy barrier that reactants pass through the minimum energy path to reach products, corresponding to the saddle point on the potential energy surface (PES).

REVIEWER COMMENTS

Reviewer #1 (Remarks to the Author):

The authors have done a reasonable job in answering my previous comments. A few further comments in response:

(1) The GitHub repository given for code access does not work. I believe that, without access to the code, this approach would not be extremely challenging to reproduce. Please can the authors ensure that this repository is working?

(2) I still have some reservations about the comparison of TS diff to other methods, such as the geodesic interpolation of Martinez, IDPP, and so on. The authors have argued that their approach generates an ensemble of conformations for a given reaction, and so is not comparable to methods like IDPP, which generate a single reaction-path approximation for a given set of input reactant/products. But the entire point of these TS-generation methods is surely to use them as part of workflows studying reactive chemistry (i.e. TS finding for input product/reactant conformers) - in other words, in order to demonstrate the utility of these schemes to the wider community, the authors need to be able to demonstrate that TSdiff can be used to identify the correct TS structures for a set of benchmark reactions (of which there are many previously studied).

However, it seems that TSdiff does not necessarily help in dealing with this challenge (as also noted by another reviewer) - instead, TSdiff generates a large number of conformers, admittedly some of which can be clustered. So - can the authors comment or test their approach in actually finding TSs for existing benchmark datasets used by other researchers who are interested in TS-finding methods? If the authors can't use TSdiff to identify TS structures for well-known benchmarks, then that draws into question how useful this approach could be to the wider community - demonstrating this would be a big boost to the impact of this paper.

Reviewer #2 (Remarks to the Author):

The authors have carefully considered all questions/comments of this reviewer, therefore this reviewer recommends publication of the article in the current form.

Reviewer #3 (Remarks to the Author):

I appreciate the careful responses to my initial review and feel that all major points have been addressed. While I still think that there is further room for improvement in evaluating the models, the authors' detailed responses are satisfying and address my concerns. The additional analysis on multiple reaction pathways is particularly helpful and provides more insight into the generative aspect of the method. I am happy to recommend the article for publication.

Response to Review 1.

We thank the reviewer for the careful evaluation and valuable comments that have helped us improve the quality of our manuscript. Our point-by-point responses are attached below, and the manuscript has been revised accordingly.

Detailed comments

1. The GitHub repository given for code access does not work. I believe that, without access to the code, this approach would not be extremely challenging to reproduce. Please can the authors ensure that this repository is working?

Reply:

The decision to keep the GitHub repository private until the publication decision was made was a deliberate choice. However, we acknowledge that it may have caused some confusion. We are pleased to announce that we have now made the GitHub repository public so that users can access and work with our code.

2. I still have some reservations about the comparison of TS diff to other methods, such as the geodesic interpolation of Martinez, IDPP, and so on. The authors have argued that their approach generates an ensemble of conformations for a given reaction, and so is not comparable to methods like IDPP, which generate a single reaction-path approximation for a given set of input reactant/products. But the entire point of these TS-generation methods is surely to use them as part of workflows studying reactive chemistry (i.e. TS finding for input product/reactant conformers) - in other words, in order to demonstrate the utility of these schemes to the wider community, the authors need to be able to demonstrate that TSDiff can be used to identify the correct TS structures for a set of benchmark reactions (of which there are many previously studied).

Reply:

We fully agree with the significance of finding the correct TS with the lowest barrier. To provide convincing results for the reviewer's concern, we performed an additional experiment on Birkholtz and Schlegel's benchmark set [1], which consists of chemical reactions commonly used to assess conventional TS discovery methods. In this experiment, we focused on the applicability of TSDiff in conjunction with a clustering method to identify the reference TS geometry. This is based on the clustering character of the geometries generated by TSDiff, as discussed with Figure 4 in the main text.

In our experiments, we performed the following steps for each reaction: First, we generated one hundred samples using TSDiff. Then, we constructed clusters based on similarities in their conformations using Ward's method [2]. Finally, we randomly selected up to two samples from each cluster and performed TS optimization on them.

We confirmed that the TS conformations corresponding to the reference geometry were found for most of the reactions tested. This result demonstrates the usefulness of TSDiff in identifying the correct TS, and we believe that TSDiff can be a valuable tool in the workflow of studying reactive chemistry.

We appreciate again for bringing this to our attention. We hope that these additional experiments will address the reviewer's concerns.

References:

- [1] Birkholz, A. B., & Schlegel, H. B. (2015). Using bonding to guide transition state optimization. *Journal of Computational Chemistry*, 36(15), 1157-1166.
- [2] Ward Jr, J. H. (1963). Hierarchical grouping to optimize an objective function. *Journal of the American statistical association*, 58(301), 236-244.

Revision:

1st paragraph in page 7 of main text.

We also present an illustrative experiment in the Supplementary Information, showcasing the practical application of a clustering algorithm in TS exploration using TSDiff.

"Application to Birkholz and Schlegel's benchmark" section in Supplementary Information.

We demonstrate a practical application of TSDiff to identifying the most energetically favorable TS, in conjunction with a clustering algorithm. To evaluate our approach, we used a benchmark set created by Birkholz and Schlegel, which consists of chemical reactions commonly used to assess conventional TS discovery methods.

...(omission of further details).

REVIEWER COMMENTS

Reviewer #1 (Remarks to the Author):

I thank the authors for their additional work to their manuscript. However, I still feel that they are missing the point of my 2nd comment about comparison to other methods.

To be more concrete, for TS-finding methods such as TSdiff, what users really want to know is how reliable it is and how efficient it is in locating the correct TS for a given reaction. Consider the recent paper "Improved Elastic Image Pair Method for Finding Transition States, Yangqiu Liu, Hexiang Qi, and Ming Lei, J. Chem. Theory Comput. 2023, 19, 2410–2417" - in that paper, the authors present a table showing how many force and Hessian evaluations are required to identify the correct TS for a benchmark set of reactions, showing the user how their proposed method performs.

In the current paper, the authors show that the TS structures can be found for some organic reactions, but as far as I can see there is no indication of efficiency or how this compares to other methods.

So, my own feeling is that this is an interesting approach reported in an interesting paper - but I don't feel that it necessarily benchmarks this approach as a route to finding TS structures. There may be reasons that this direct comparison is not appropriate - which the authors should explain if that is the case.

In any case, I'm happy for this to be taken as an editorial decision.

Response to Review 1.

We thank the reviewer for the careful evaluation and valuable comments that have helped us improve the quality of our manuscript. Our detailed responses of the reviewer's comments are attached below, and the manuscript has been revised accordingly.

Detailed comments

Comments:

I still feel that they are missing the point of my 2nd comment about comparison to other methods.

To be more concrete, for TS-finding methods such as TSDiff, what users really want to know is how reliable it is and how efficient it is in locating the correct TS for a given reaction. Consider the recent paper "Improved Elastic Image Pair Method for Finding Transition States, Yangqiu Liu, Hexiang Qi, and Ming Lei, J. Chem. Theory Comput. 2023, 19, 2410–2417" - in that paper, the authors present a table showing how many force and Hessian evaluations are required to identify the correct TS for a benchmark set of reactions, showing the user how their proposed method performs.

In the current paper, the authors show that the TS structures can be found for some organic reactions, but as far as I can see there is no indication of efficiency or how this compares to other methods.

Reply:

We acknowledge the importance of comparative study to validate the reliability and efficiency of our method, and we appreciate your valuable input.

For the efficiency evaluation, the direct comparison with other TS-finding methods was limited due to its complexity which arises from the uniqueness of TSDiff. The conventional methods, including IDPP, GI, and EIP, are designed to find the "corresponding" TS given specific reactant and product geometries. It should be noted that preparing the appropriate reactant and product geometries entail substantial computational costs. In contrast, TSDiff can give multiple TSs without knowing the structural information of reactants and products, thanks to its generative nature and deep learning power. Consequently, TSDiff can save almost all the costs of preparing reactant and product geometries.

In response to the reviewer's request, therefore, we have included a cost analysis based on the number of force calls and Hessian calls throughout the entire TS search process starting with initial TSs given by TSDiff in the "S.2 Application to Birkholz and Schlegel's benchmark" section. In addition, we have appended a cost comparison with the EIP method, focusing only on the generated TS that converges to the exact reference TS. All of the cost analyses are performed on Birkholz and Schlegel's benchmark. Notably, our results reveal that around 14 force calls with one Hessian call are required per a reaction for post-optimization.

Revision:

The 1st paragraphs in the section "3. Discussion" (page 15 of Manuscript)

...Furthermore, TSDiff's transferability is supported by its successful application to another benchmark dataset, as described in the "Application to Birkholz and Schlegel's benchmark" section of the Supplementary Information. Here, TSDiff also proves to be an effective initial TS guesser, requiring only a small number of force calls during subsequent TS optimization. Therefore, this study shows the promising potential of TSDiff for efficient and reliable TS exploration.

The 5th and 6th paragraphs in the section "S.2 Application to Birkholz and Schlegel's benchmark" (page 5 of Supporting Information)

To illustrate the efficiency of the TS search process using TSDiff, we present the number of force calls in the TS optimization based on the Berny algorithm in Table S1. Furthermore, in Table S2, we compare the number of force and Hessian calls with those of the EIP [11] and i-EIP [12] methods, which are recently introduced TS finding methods based on a double-ended approach. Both methods utilize atomic forces and Hessians obtained by quantum calculations with reactant and product geometries as input to locate the TS geometry. The values for EIP and i-EIP in the table are sourced from their original paper [12], which employed density functional theory calculations at the B3LYP-D3/6-31G* level. The results for the MeOH reaction have been omitted as they are not available in the original paper. For a fair comparison for the same reaction conformer, the TS optimization results corresponding to the reactions for which TSDiff found the reference TS conformation are included in the comparison. Finally, the results are compared across a total of 10 reactions, as shown in Table S2.

TSDiff consistently produced the smallest number of force calls for all reactions listed in Table S2. While EIP and i-EIP require reactant and product geometries as input and are designed to locate a specific TS conformation, TSDiff generates multiple TS conformations. Hence, it can be challenging to make an equivalent efficiency comparison between them. However, the remarkably low number of force calls required by TSDiff clearly demonstrates its efficiency in TS optimization. Consequently, these results confirm that TSDiff can simplify demanding tasks in input preparation, such as establishing molecular conformations and orientations of reactant and product, while significantly improving the efficiency of the subsequent optimization process through its precise geometry generation.

Table S1 in the section "S.2 Application to Birkholz and Schlegel's benchmark" (page 4 of SI)

Table S1: TS optimization results based on TS conformers generated by TSDiff. For each reaction in Birkholz and Schlegel's benchmark, the number of clusters and optimized TSs obtained, whether the reference TS conformation is included in the optimized TSs, and the number of force calls in the TS optimization computations averaged over clusters are shown.

Table S2 in the section "S.2 Application to Birkholz and Schlegel's benchmark" (page 6 of SI)

Table S2: Comparison of TSDiff, EIP, and i-EIP methods. For the reactions in Birkholz and Schlegel's benchmark, the number of force and Hessian calls of TS optimization computations are compared. The values of EIP and i-EIP are borrowed from [12].

REVIEWERS' COMMENTS

Reviewer #1 (Remarks to the Author):

The additions and related explanation help address my previous comment satisfactorily at this point - I have nothing further to add.

For future demonstrations of this method, it may be useful to further consider how the efficiency and applicability - relative to existing methods - could be evaluated.

Response to Review 1.

We thank the reviewer for the careful evaluation and valuable comments that have helped us improve the quality of our manuscript.

As mentioned by the reviewer, we believe that research on how to compare the TSDiff model with other TS locating methods is still needed. In particular, we believe that research on TS geometry location method using only 2D connectivity information will be actively pursued starting from this study, and we expect it to be addressed in future studies.